# Comparison of the performances of survival analysis regression models for analysis of conception modes and risk of type-1 diabetes among 1985–2015 Swedish birth cohort

**Adeniyi Francis Fagbamigbe**[1,2,3]*, **Emma Norrman**[4,5], **Christina Bergh**[4,5], **Ulla-Britt Wennerholm**[4,5], **Max Petzold**[6]

**1** Department of Epidemiology and Medical Statistics, College of Medicine, University of Ibadan, Ibadan, Nigeria, **2** Health Data Science Group, Division of Population and Behavioural Sciences, School of Medicine, University of St Andrews, St Andrews, United Kingdom, **3** Populations, Evidence and Technologies Group, Division of Health Sciences, University of Warwick, Coventry, United Kingdom, **4** Department of Obstetrics and Gynecology, Institute of Clinical Sciences, Sahlgrenska Academy, University of Gothenburg, Gothenburg, Sweden, **5** Sahlgrenska University Hospital/Östra, Gothenburg, Sweden, **6** School of Public Health and Community Medicine, Institute of Medicine, University of Gothenburg, Gothenburg, Sweden

* franstel74@yahoo.com, fadeniyi@cartafrica.org

## Abstract

The goal is to examine the risk of conception mode-type-1 diabetes using different survival analysis modelling approaches and examine if there are differentials in the risk of type-1 diabetes between children from fresh and frozen-thawed embryo transfers. We aimed to compare the performances and fitness of different survival analysis regression models with the Cox proportional hazard (CPH) model used in an earlier study. The effect of conception modes and other prognostic factors on type-1 diabetes among children conceived either spontaneously or by assisted reproductive technology (ART) and its sub-groups was modelled in the earlier study. We used the information on all singleton children from the Swedish Medical Birth Register hosted by the Swedish National Board of Health and Welfare, 1985 to 2015. The main explanatory variable was the mode of conception. We applied the CPH, parametric and flexible parametric survival regression (FPSR) models to the data at 5% significance level. Loglikelihood, Akaike and Bayesian information criteria were used to assess model fit. Among the 3,138,540 singletons, 47,938 (1.5%) were conceived through ART (11,211 frozen-thawed transfer and 36,727 fresh embryo transfer). In total, 18,118 (0.58%) of the children had type-1 diabetes, higher among (0.58%) those conceived spontaneously than the ART-conceived (0.42%). The median (Interquartile range (IQR)) age at onset of type-1 diabetes among spontaneously conceived children was 10 (14–6) years, 8(5–12) for ART, 6 (4–10) years for frozen-thawed embryo transfer and 9 (5–12) years for fresh embryo transfer. The estimates from the CPH, FPSR and parametric PH models are similar. There was no significant difference in the risk of type-1 diabetes among ART- and spontaneously conceived children; FPSR: (adjusted Hazard Ratio (aHR) = 1.070; 95% Confidence Interval (CI):0.929–1.232, p = 0.346) vs CPH: (aHR = 1.068; 95%CI: 0.927–1.230, p = 0.361). A sub-analysis showed that the adjusted hazard of type-1 diabetes was 37% (aHR = 1.368;

**Data Availability Statement:** Ethical permission was given from the Regional Ethical Committee (bjorn.rydevik@gu.se) at the University of Gothenburg (Dnr 214-12, T422-12, T516-15, T233-16, T300-17, T1144-17, T121-18). The Swedish National Board of Health and Welfare obtained informed written consent from patients to have data/samples from their medical records used in research. We followed the prescribed ethical regulations on confidentiality when the database was accessed in October 2019. The Swedish National Board of Health and Welfare, and Statistics Sweden (SCB) placed ethical restrictions on sharing the data publicly as the data contain potentially identifying or sensitive patient information. The restriction was concurred to by the Regional Ethical Committee at the University of Gothenburg, Sweden.

**Funding:** The author(s) received no specific funding for this work.

**Competing interests:** The authors have declared that no competing interests exist.

**Abbreviations:** AFT, Accelerated Failure Time; AIC, Akaike information criteria; aHR, adjusted Hazard Ratio; ART, assisted reproductive technology; BIC, Bayesian information criteria; CI, Confidence Interval; CPH, Cox Proportional Hazard; FPSR, Flexible parametric survival regression; ICSI, Intracytoplasmic sperm injection; IQR, Interquartile range; IVF, In-Vitro-Fertilization; PH, Proportional hazard; RP, Royston-Parmar; SB, Statistics Sweden; SC, Spontaneous conception.

95%CI: 1.013–1.847, p = 0.041) higher among children from frozen-thawed embryo transfer than among children from spontaneous conception. The hazard of type-1 diabetes was higher among children whose mothers do not smoke (aHR = 1.296; 95%CI:1.240–1.354, p<0.001) and of diabetic mothers (aHR = 6.419; 95%CI:5.852–7.041, p<0.001) and fathers (aHR = 8.808; 95%CI:8.221–9.437, p<0.001). The estimates from the CPH, parametric models and the FPSR model were close. This is an indication that the models performed similarly and any of them can be used to model the data. We couldn't establish that ART increases the risk of type-1 diabetes except when it is subdivided into its two subtypes. There is evidence of a greater risk of type-1 diabetes when conception is through frozen-thawed transfer.

## Introduction

An exploratory analysis of age at onset of type-1 diabetes among children born in Sweden between 1985 and 2015 and conceived either spontaneously or by assisted reproductive technology (ART) showed that (i) skewed discrete age distribution because most ART-conceived children are younger (ART is increasing and most children were born in the more recent years), (ii) a probable peak in the risk of type-1 diabetes at 10–14 years of age, (iii) there is an increased risk of developing diabetes in more recent years [1]. The ART was either the standard in-vitro-fertilization (IVF) or intracytoplasmic sperm injection (ICSI) with fresh or frozen-thawed embryo transfer. This study used the Cox proportional hazard (CPH) model to evaluate if the mode of conception (ART vs spontaneous) was associated with an increased risk of type-1 diabetes [1]. However, it is not known whether this method captured the peculiarities of the data. There is a need to investigate if the differences were caused by skewness in age, having more ART-conceived children during the later years when diabetes is more prominent.

There have been contradictions in the reported risk of conception mode-type-1 diabetes in the literature [1–3]. Norrman et al. reported an insignificant risk of Type-1 diabetes by maternal fertility (adjusted hazard ratio (aHR) = 1.07; 95% CI: 0.93–1.23) but significant differences in risk were found between frozen embryo transfer and type-1 diabetes (aHR 1.52; 95% CI: 1.08–2.14 and 1.41; 95% CI: 1.05–1.89) for frozen versus fresh and frozen versus spontaneous conception (SC), respectively in a subgroup analyses of a Swedish data [1]. Using Danish data, Hargreave et al. found that the risk of type-1 diabetes was not significant with conception mode (aHR = 1.01; 95% CI: 0.90–1.13) after adjusting for other covariates [3]. Similarly, Kettner et al. reported no association between fertility treatment and childhood type-1 diabetes mellitus but found that ovulation induction or intrauterine insemination with follicle-stimulating hormone was associated with a higher risk of type-1 diabetes mellitus (aHR = 3.22; 95% CI: 1.20 to 8.64) [2]. The authors found no association with diabetes type-1 diabetes in other types of fertility treatment indications [2]. The data analysed by Hargreave et al. and Kettner et al. were both Danish but differed in sizes with over 1, 550,519 and 565,116 births respectively whereas there were 3,138,540 births in the study by Norrman et al. [1–3]. The study setting, sample sizes and methods of analysis might have caused these contradictions. The reason for these contradictory findings could be that the analyses used in the prior literature depended on the appropriateness of proportional hazards models in the settings. This paper is therefore designed to revisit and assess the effect of ART and its subgroups, on the development of type-1 diabetes. To achieve this, we assessed the fit of a range of different survival analysis regression

models in assessing the effect of ART in having type-1 diabetes among a birth cohort in Sweden.

The data used in Norrman et al. presented right-censored data on the onset of type-1 diabetes which makes survival analysis a natural choice of analysis [1]. Survival analysis, also called analysis of time-to-event, is a statistical procedure for a follow-up study in which some subjects may not have experienced the event of interest due to loss to follow up, withdrawal of subjects from the study or the study coming to an end [4, 5]. Besides the estimation of the survival function of such time-to-event, greater interest is the identification of prognostic factors that affects the timings [6–9]. Several methodologies, known as survival regression methods, have been developed to achieve this goal. While some of the models are traditional survival regression models, some are modern [10, 11]. The traditional survival regression models consist of the non-parametric, semi-parametric and parametric models. We describe the assumptions, advantages and disadvantages of these models in the next paragraph.

The Kaplan-Meier estimator is the nonparametric maximum likelihood estimate (NPMLE) of the survivor function when the model is a non-parametric survival model. It does not allow the investigation of the effects of covariates [5]. Hence the effects of risk factors on the time-to-event cannot be estimated. Nonetheless, the Kaplan-Meier estimator allows the exploration of the effect of covariates graphically and by comparison of distinct estimates of the survival function but this gets cumbersome when several covariates are to be investigated together. The CPH model is a semi-parametric regression model and it is the most commonly used survival analysis regression models [12–16]. This model has a unique advantage of not making any assumptions about the shape of the underlying hazard function and can directly use the hazard ratios to make a reliable estimation of the treatment effects [14] but rather assumes that the hazard ratios of the covariates in the model are constant over time [16, 17]. Although the CPH model is very useful and often accurate for a large sample size study, the major criticism against the CPH models is that this assumption is often violated, especially in a very long follow-up study. The CPH model has another disadvantage in that the survival function hence the cumulative hazard function, are continuous with respect to Lebesgue measure [18]. The estimators are, however, step functions rather than being smooth [11, 19].

The parametric models assume that the survival and hazard functions follow a specific distribution in which case the parameters can be estimated. The parametric survival models, such as the exponential and Weibull models in addition to the log-logistic and generalized gamma models are often used to obtain smooth hazard rates and cumulative hazard functions of the risk factors and to extrapolate the survival functions and cumulative functions [5, 20]. A major disadvantage of the parametric models lies in their insufficient flexibility to ensure adequate representation of the data been fitted, and the fact that the cumulative hazard function or survival function may be biased [19]. These inflexibilities often lead to biased estimates for the cumulative hazard and/or survival functions. Moreover, parametric models with complex underlying hazard fail to capture the true (covariate) effects [21, 22].

The Flexible parametric survival regression (FPSR) models were developed by Royston and Parmar in the early 2000s [23] as a more flexible alternative. The FPSR attempted to eliminate the shortcoming of the CPH and the parametric models by building on the strengths of these models. The flexible models are an extension of several parametric survival models [23] and were formulated to relax the assumption of linearity of log time [24]. These flexible models represent the log of the baseline cumulative hazard function with a restricted cubic spline function of the logarithm function of time [4, 21, 24]. The literature is replete with several advantages of the FPSR models [4, 5, 14, 21, 22, 25, 26].

This study is a method paper with an application. The analyses were carried out to assess the effect of ART on the development of type-1 diabetes and compare the performance of the

CPH, parametric and the FPSR models in the identification and quantification of the hazards of prognostic factors of the timing of onset of type-1 diabetes among ART (including fresh and frozen embryo transfer) and spontaneously conceived children in Sweden between 1985 and 2015. We ascertained the effect of the differences in age distribution between ART and spontaneously conceived children on the risk of having type-1 diabetes. In this study, we applied and compared the fit and performances of (i) Cox proportional hazard (CPH) models, (ii) The proportional hazard (PH) (Weibull PH, exponential PH and Gompertz PH) and the Accelerated Failure Time (AFT) (Weibull AFT, exponential AFT, log-logistic, lognormal, and the generalized gamma models) parametric survival models and (iii) Royston-Parmar (RP) Flexible parametric survival regression (FPSR) models with a different number of knots. The assumption of the PH can be relaxed in the PH models. The FPSR models have the advantage that the PH assumption can be relaxed rather easily in comparison to the other survival models.

## Methods

### The data

The data used for this study were collected in Sweden and represent all singleton children (n = 3,138,540) born between 1985 and 2015, excluding singletons conceived after oocyte donation. The data was extracted from the national health data registers hosted by the Swedish National Board of Health and Welfare, linked with several national quality registers and information from Statistics Sweden (SCB). The SCB maintains a constantly updated and quality-checked register from the Swedish National Board of Health and Welfare, and the Swedish National Data Service which ensures the completeness of the data. The national quality registers include registers on morbidity and mortality of children born in Sweden. The SCB did not state any change in the inclusion criteria of morbidities and mortalities over time.

Further details on the data have been reported [1]. Ethical permission was given from the Regional Ethical Committee (bjorn.rydevik@gu.se) at the University of Gothenburg (Dnr 214–12, T422-12, T516-15, T233-16, T300-17, T1144-17, T121-18). The Swedish National Board of Health and Welfare obtained informed written consent from patients to have data/ samples from their medical records used in research. We followed the prescribed ethical regulations on confidentiality when the database was accessed in October 2019. The Swedish National Board of Health and Welfare, and Statistics Sweden (SCB) placed ethical restrictions on sharing the data publicly as the data contain potentially identifying or sensitive patient information. The restriction was concurred by the Regional Ethical Committee at the University of Gothenburg, Sweden.

### Statistical models

**Cox proportional hazard models.**   The CPH model was developed based on the assumption that hazards are multiplicatively proportional to baseline hazards [27] but without any assumption about the distribution of the hazards.

$$h(t) = h_0(t)e^{\beta_1 x_1 + \beta_2 x_2 + \cdots \beta_k x_k} \tag{1}$$

From Eq (1), coefficients $\beta_1, \beta_2, \ldots \ldots, \beta_k$ could be estimated but a direct estimate of the baseline hazard ($h_0(t)$) or its distribution cannot be estimated. The model nonetheless provides an avenue to estimate the baseline cumulative hazard ($H_0(t)$) and baseline survival ($S_0(t)$) which can be used to estimate the $h_0(t)$ [19].

The CPH uses maximum partial likelihood methods to estimate coefficients ($\beta_i$). Numerically, let $x_i$ be the row vector of covariates for the time interval ($t_{0i}$; $t_i$) for the $i^{th}$ observation in

a dataset with N subjects (i = 1, 2, 3,. … … … .., N). The coefficient ($\beta_i$) of the covariates ($X_i$) can be estimated by maximizing the partial log-likelihood function

$$logL = \sum_{j=1}^{P} \left[ \sum_{i \in P_j} x_i\beta - d_j \, log \left\{ \sum_{k \in R_j} exp(x_k\beta) \right\} \right] \tag{2}$$

where j indexes the ordered failure times $t_{(j)}$, j = 1, 2, …… …, P; $P_j$ is the set of $P_j$ observations that fail at $t_{(j)}$; $d_j$ is the number of failures at $t_{(j)}$; and $R_j$ is the set of observations k that are at risk at time $t_{(j)}$ (that is, all k such that $t_{(0k)} < t_{(j)} \leq t_{(k)}$).

**Parametric survival models.** As reported in the literature [5, 20], the accelerated failure-time (AFT) models and the multiplicative or proportional hazards (PH) model are the most-used parametric models for adjusting survivor functions for the effects of covariates. The PH models include the Weibull, Gompertz, and exponential while the exponential, Weibull, log-normal, log-logistic, and generalized gamma are the commonest AFT models. In the AFT model, log(t) (the natural logarithm of the survival time) is expressed as a linear function of the covariates which yields a linear model

$$logt_j = X_j\beta + z_j. \tag{3}$$

where $x_j$ is a vector of the covariates, $\beta$ is a vector of the regression coefficients, and $z_j$ is the error function whose distributional form determines the form of the regression model. For instance, a normal density leads to the lognormal regression model just as the logistic density yields the log-logistic regression. The AFT models applied to change the time scale by a factor of $e^{-x_j\beta}$. The time is accelerated if the factor is <1 and decelerated (degraded) if >1. In the PH model, the concomitant covariates have a multiplicative effect on the hazard function and contain the covariates which have a multiplicative effect on the hazard function

$$h(t_j) = h_0(t)g(X_j) \tag{4}$$

for some $h_0(t)$ and for $g(X_j)$, a non-negative covariates function. In some cases, $g(X_j)$ is expressed as $e^{x_j\beta}$. The parametric PH model becomes a CPH model if the function $h_0(t)$ is not specified [5, 20].

**Flexible parametric survival regression model.** These models use restricted cubic splines to model a transformation of the survival function and can be modelled on different scales, including the hazard scale, the odds scale, and the probit scale in flexible parametric survival analyses. We focussed on the hazard scale of the models to ensure that the estimates from the FPSR and CPH models are comparable. This approach has an advantage in that the corresponding function is more stable and the process of capturing the shape of the function is easier [28]. The Weibull model is one of the commonest parametric models for survival data and can be criticized due to its lack of flexibility in the shape of the hazard function. The log cumulative hazard function of Weibull distribution is written as

$$\ln \{H(t : x)\} = \ln(H_0(t)) + x_i\beta_i = \ln \lambda + \gamma \ln(t) + x_i\beta_i \tag{5}$$

In Eq (5), $\ln H_0(t)$ is the baseline log cumulative hazard at time point t, $\lambda$ is the scale parameter, $\gamma$ is the shape parameter, $x_i$ is the covariate and $\beta_i$ is the coefficient of the covariate [28]. The log cumulative hazard function of the FPSR models are based on a log cumulative hazard scale with the use of restricted cubic spline function of the log time to transform the function

in Eq (5) is

$$\ln\{H(t:x)\} = \ln H_0(t) + x_i\beta_i = s(x) + x_i\beta_i \tag{6}$$

In Eq (6), $x = \ln(t)$ and $s(x)$ is a restricted cubic spline function with k knots $k_1$, $k_2, \ldots\ldots\ldots\ldots.k_k$ and parameters $\gamma_0, \gamma_1, \ldots\ldots\ldots\ldots..\gamma_{k-1}$ is expressed as

$$s(x) = \gamma_0 + \gamma_1 z_1 + \cdots\ldots\ldots\ldots + \gamma_{k-1} z_{k-1}$$

Where $z_1 = x = \ln(t)$ and $z_j(j \geq 2)$ are derivable functions that are determined by the numbers and the positions of the knots is computed as

$$z_j = (x - k_j)_+^3 - \phi_j(x - k_j)_+^3 - (1 - \phi_j)(x - k_j)_+^3 \ j = 2, \ldots, k - 1$$

Where $\phi_j = (k_k - k_j)/(k_k - k_1)$ [4, 19, 23].

The desired complexity of the FSPR models is specified as the number and positions of the knots (connection points) in log time of the spline's cubic polynomial segments [19]. The $k$ knots, a maximum of 9 knots, has $k+1$ degrees of freedom (df). The placement of knots is at the centiles (computed as *100/df*) of event times. For 4 Knots, the df is 5 and the knots will be located at centiles 20, 40, 60 and 80. The internal knots are bounded by the "boundary knots" which are placed at the minimum and maximum of the distribution of uncensored survival times. The FSPR model reduces to the Weibull model if the number of knots is 0, with $\gamma_0$ and $\gamma_1$ as the estimates of its scale and shape parameter respectively. Royston et al. suggested either 1 or 2 knots for smaller ($<$10,000) datasets and 4 or 5 knots for larger ($>$ = 10,000) datasets [19]. The FSPR model was implemented in Stata using "stpm2" command. The stpm2 can be used with single- or multiple-record or failure survival data on the log cumulative hazard, the log cumulative odds, the probit scales, or on a scale based on theta-value using the Aranda-Ordaz family of link functions [9, 24]. We computed the incidence rate as the probability of having diabetes per person-year and the attributable fraction to assess the excess risk attributable to been conceived with ART compared with spontaneous conception. The survival analysis was performed using Stata 16.0 (Stata-Corp LP, College Station, TX, USA). Microsoft Excel in Office 365 and R Software were used for data visualization.

**Dependent variable.** The outcome is a paired survival time and indicator of whether the participant had experienced type-1 diabetes or not. For those that had type-1 diabetes, the survival time was their age as of its onset, while the time (age) to type-1 diabetes was censored on the date of the data collection, emigration and death for those with no type-1 diabetes.

**Independent variable.** Our choice of covariates was informed by Norman et al. [1]. The main explanatory variable in this study is the mode of the conception of the children: spontaneous or ART conception. The types of ART considered in this study are IVF and ICSI with fresh and frozen-thawed embryo transfer. The other independent variables are sex of the child, maternal age, parity, year of birth (birth cohort), mother's and father's country of birth, mothers' smoking status, mothers' and fathers' diabetic status, and maternal and paternal education at the birth of the child.

**Model selection criteria.** The log-likelihood, Akaike information criteria (AIC) [29] and the Bayesian information criteria (BIC) [30] were used to assess the fit of the models. The lower these quantities the better the fitness of the models. The AIC and the BIC are usually computed and compared separately among different models to determine the best fitting model. In all cases, the lower the AIC and BIC, the better the model [14]. Literature suggests that AIC will choose a more complex model irrespective of sample size while BIC is more likely to choose a simpler model [14]. AIC is often preferable in situations when a false negative finding would be considered to be more misleading than a false positive, and BIC is better in

situations where a false positive is as misleading as, or more misleading than, a false negative [14]. We used these criteria to assess which of the models been compared was the optimal model.

## Results

Among the 3,138,540 children included in the analysis, 98.5% (3,090,602) were conceived spontaneously while 1.5% (47,938) were conceived through ART. Of the 47,938 ART-conceived children, 36,727 were through fresh embryo transfer and 11,211 were from frozen-thawed embryo transfer. About 51.4% of the children were males, nearly half (47.8%) were from mothers aged 30–39 years, 81.7% were from Swedish mothers, from mothers (0.4%) and fathers (0.6%) who were diabetic. In total, 18,118 (0.58%) became type-1 diabetic during the follow-up period comprising 50,936,586 person-years at risk. Among those conceived spontaneously, 0.58% (n = 17,916) became diabetic compared with 0.42% (n = 202) among the ART-conceived (Table 1). The median (Interquartile Range (IQR)) age at onset of type-1 diabetes among spontaneously conceived children was 10 (6–14) years, 8 (5–12) for ART, 9 (5–12) years for fresh embryo transfer and 6 (4–10) years for frozen-thawed embryo transfer. Children conceived by ART had a higher incidence rate of type-1 diabetes compared with those conceived spontaneously. The overall incidence rate of those conceived through ART and spontaneously were 43 and 36 per 100 000 person-years at risk respectively. The highest incidence of type-1 diabetes was found among children whose fathers (320/100,000) and mothers (244/100,000) were diabetic compared with those whose parents were not diabetic (34/100,000 and 35/100,000). The distribution of the incidence of type-1 diabetes by selected children characteristics is presented in Fig 1.

### Test of equality of incidence rates of type-1 diabetes

The overall difference in unadjusted incidence rate between the children conceived spontaneously and by ART was 0.000791 (95% CI: 0.000019–0.000139, incidence rate ratio was 1.222589 (95% CI: 1.058–1.40448), attributable fraction exposed was 0.18206 (95% CI: 0.05568–0.28799). The two tail hypothesis test that the incidence rates for each of the groups are not different was significant (p = 0.0058) (Not shown in the Tables).

### Distribution of the hazard and survival functions under different distributions

On the choice of the survival regression analysis to be adopted, we first explored the CPH model. As shown in Fig 2 (left panel), the assumption of proportionality was not violated. In the same way, a test of non-proportionality of Kaplan Meier survival curves of the observed and predicted ART estimates at various ages of the children was not significant (Fig 2: right panel).

In Fig 3, we present the survival and hazard functions of developing type-1 diabetes by modes of conception using the Kaplan Meier, smoothed, CPH and the FSPR models. In all, survivorship was higher among the spontaneously conceived children compared with those conceived through ART. In the same way, the hazard of type-1 diabetes was higher among those conceived through ART compared with those that were spontaneously conceived.

In Fig 4, we compared the hazard and survival functions of the flexible models at different degrees of freedom (knots/splines). The functions appear more realistic and flexible at higher degrees of freedom than the Weibull distribution. The greatest flexibility was noticeable at df = 6. The lower panel of Fig 4 shows the predicted incidence of diabetes using the flexible model.

**Table 1. Distribution of children' characteristics and incidence rate of type-1 diabetes.**

| Characteristics | n (%) | Prevalence (%) | *Median (IQR) years Age | **Incidence per 100,000 |
|---|---|---|---|---|
| Conception Mode | | | | |
| Spontaneous | 3,090,602 (98.5) | 0.58 | 10(6–14) | 36 |
| ART | 47,938 (1.5) | 0.42 | 8(5–12) | 43 |
| Fresh^ | 36.727(1.2) | 0.43 | 9(5–12) | 41 |
| Frozen-thawed^ | 11,211(0.3) | 0.40 | 6(4–10) | 53 |
| Sex | | | | |
| Male | 1,613,900 (51.4) | 0.62 | 11(6–15) | 38 |
| Female | 1,524,640 (48.6) | 0.53 | 9(6–13) | 33 |
| Birth Cohort | | | | |
| 1985–1990 | 634,215 (20.2) | 0.80 | 13(9–19) | 29 |
| 1991–1995 | 554,632 (17.7) | 0.84 | 12(7–16) | 36 |
| 1996–2000 | 426,536 (13.6) | 0.80 | 10(6–14) | 44 |
| 2001–2005 | 463,312 (14.8) | 0.61 | 9(5–11) | 46 |
| 2006–2010 | 518,879 (16.5) | 0.32[a] | [a]5(3–7) | 36 |
| 2011–2015 | 540,966 (17.2) | 0.09[a] | [a]4(2–3) | 24 |
| Mother age at birth | | | | |
| <20 | 44,917 (1.4) | 0.55 | 11(7–16) | 30 |
| 20/24 | 465,164 (14.8) | 0.61 | 11(7–16) | 33 |
| 25/29 | 1,017,464 (32.4) | 0.62 | 10(6–15) | 36 |
| 30/39 | 1,501,187 (47.8) | 0.55 | 10(6–14) | 36 |
| 40+ | 109,808 (3.5) | 0.50 | 9(5–12) | 37 |
| Parity | | | | |
| First | 1,344,674 (42.8) | 0.57 | 10(6–14) | 36 |
| Multi | 1,793,866 (57.2) | 0.58 | 10(6–14) | 35 |
| Smoking mother | | | | |
| No | 2,706,434 (86.2) | 0.57 | 10(6–14) | 37 |
| Yes | 432,106 (13.8) | 0.60 | 12(7–17) | 28 |
| Mothers' country | | | | |
| Sweden | 2,563,910 (81.7) | 0.64 | 10(6–14) | 38 |
| Nordic | 85,708 (2.7) | 0.62 | 11(7–15) | 35 |
| Europe | 173,997 (5.5) | 0.23 | 10(6–14) | 17 |
| Other | 314,486 (10.0) | 0.23 | 8(4–12) | 19 |
| Fathers' country oo ofof birthfodlandgr_ | | | | |
| Sweden | 2,526,560 (81.2) | 0.64 | 10(6–14) | 38 |
| Nordic | 79,013 (2.5) | 0.62 | 11(7–15) | 35 |
| Europe | 190,586 (6.1) | 0.23 | 10(6–14) | 17 |
| Other | 315,239 (10.1) | 0.26 | 8(5–12) | 21 |
| Mothers' education | | | | |
| < = 9 years | 303,360 (9.7) | 0.53 | 11(6–15) | 32 |
| 10–12 years | 1,377,240 (44.2) | 0.64 | 10(6–15) | 36 |
| Higher edu<3 years | 459,124 (14.7) | 0.60 | 10(6–14) | 36 |
| Higher edu> = 3 years | 974,777 (31.3) | 0.50 | 9(5–13) | 35 |
| Fathers' education | | | | |
| < = 9 years | 440,486 (14.3) | 0.58 | 11(7–15) | 31 |
| 10–12 years | 1,544,336 (50.1) | 0.63 | 10(6–14) | 37 |
| Higher edu<3 years | 440,789 (14.3) | 0.59 | 10(6–14) | 36 |
| Higher edu> = 3 years | 659,817 (21.4) | 0.47 | 9(5–14) | 34 |

(*Continued*)

**Table 1.** (Continued)

| Characteristics | n (%) | Prevalence (%) | *Median (IQR) years Age | **Incidence per 100,000 |
|---|---|---|---|---|
| Type-1 diabetes mother | | | | |
| No | 3,125,405 (99.6) | 0.56 | 10(6–14) | 35 |
| Yes | 13,135 (0.4) | 3.56 | 8(4–12) | 244 |
| Type-1 diabetes father | | | | |
| No | 3,120,677 (99.4) | 0.55 | 10(6–14) | 34 |
| Yes | 17,863 (0.6) | 4.78 | 8(5–13) | 320 |
| Total | 3,138,540 | 0.58 | 10(6–14) | 36 |

^subdivisions of ART

*median age at the onset of type-1 diabetes, IQR Interquartile Range

**Incidence per 100,000 person-year

aLimited follow-up time

## Comparison of the models

The statistics of all the model selection criteria considered in this study (the loglikelihood, the AIC and the BIC) are shown in Table 2. The AIC and BIC of the FPSR models, the statistics obtained from the parametric models including the Weibull AFT, exponential AFT, log-logistic, lognormal, and the generalized gamma models were only slightly higher than that of the FPSR models. Among the FPSR models evaluated at different degrees of freedom, the largest loglikelihood, the lowest AIC and the lowest BIC were at 6 degrees of freedom (df = 6). Although the model fits are similar for all the models, the FPSR has the lowest values for each of the criteria. This is further supported by the significance of restricted cubic splines (rcs) of each covariate otherwise called the slope of the hazard curve within each of the knots generated at 6 degrees of freedom (Table 2).

Table 3 shows the coefficients of the covariates included in the adjusted models and their respective standard errors (s.e). In all, the estimates from the CPH model, the parametric PH models and the FPSR model at different degrees of freedom were similar and generally lower than the estimates from the parametric AFT models.

## Modelling the risk factors of type-1 diabetes

The estimates from each of the model were similar. We have used the estimates from the FPSR model at 6 degrees of freedom to interpret the adjusted determinants of timing of the onset of type-1 diabetes among the children. The age of the children was used as a timescale to correct for the problem associated with lesser exposure time among the late entries. Although insignificant, the adjusted hazard of type-1 diabetes was 7% higher (adjusted Hazard Ratio (aHR) = 1.070; 95% Confidence Interval (CI): 0.929–1.232, p = 0.346)) among children conceived through ART than those conceived spontaneously while controlling for other variables. A sub-analysis with emphasis on whether the transferred embryo was fresh or frozen-thawed transferred while controlling for other covariates showed that the hazard of type-1 diabetes was 37% (aHR = 1.368; 95% CI: 1.013–1.847, p = 0.041) higher among frozen-thawed transfer compared with those conceived spontaneously. Inversely, the hazard of type-1 diabetes was higher among frozen-thawed embryo transfer than fresh embryo transfer (aHR = 1.407; 95% CI: 1.007–1.965, p = 0.046; not shown in the Tables). While controlling for other variables, the hazard of type-1 diabetes was about 17% (aHR = 1.171; 95% CI: 1.137–1.206, p<0.001) higher among males than the females. The adjusted hazard of type-1 diabetes was significantly

(a) Comparison of the incidence rate per age at onset of type-1 diabetes among children mode of conception

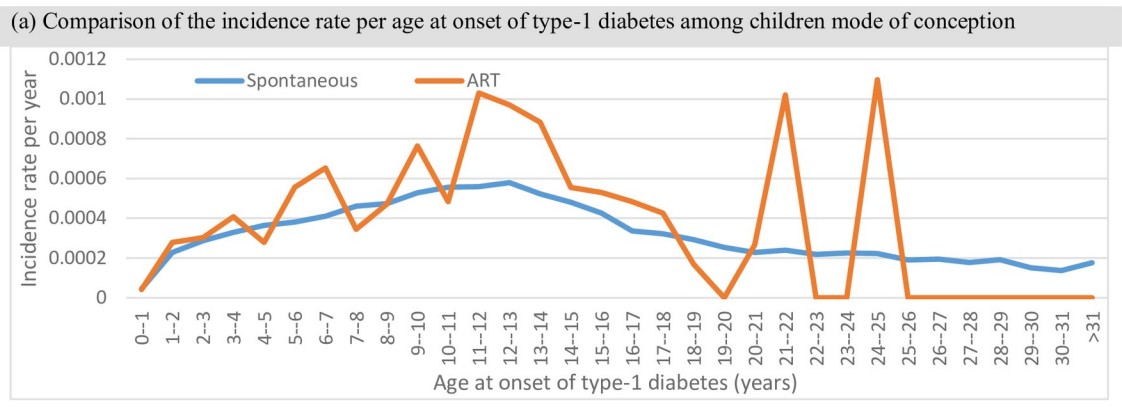

(b) Comparison of the incidence rate per 100,000 person year of type-1 diabetes by children mode of conception

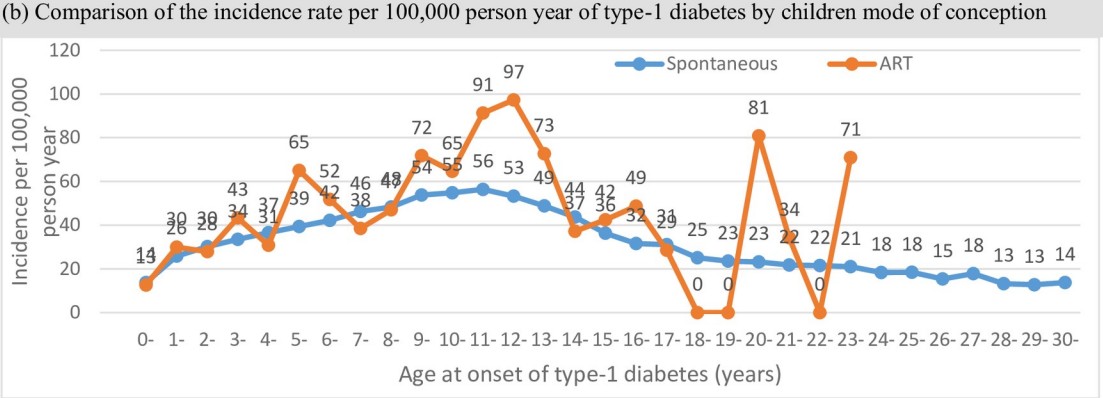

(c ) Comparison of the incidence rate per 100,000 person year of type-1 diabetes by sex and year of birth

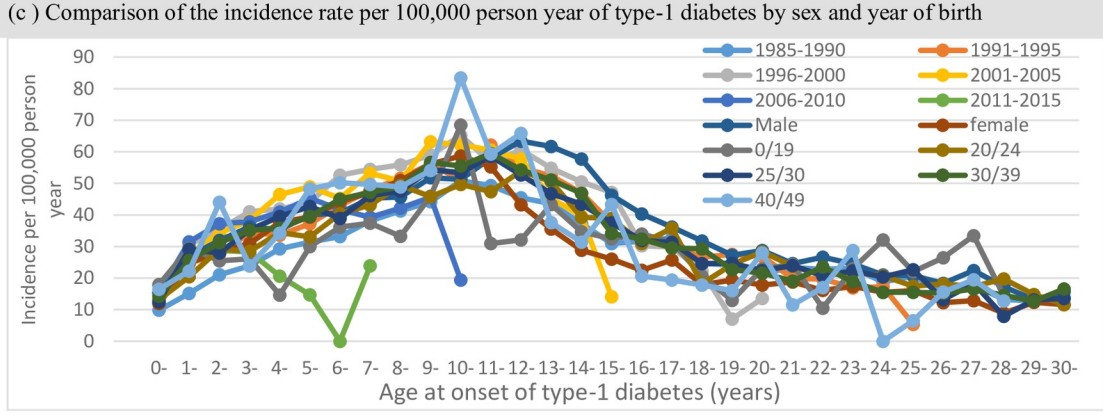

(d) Comparison of the incidence rate per 100,000 person-year of type-1 diabetes by parental diabetes status

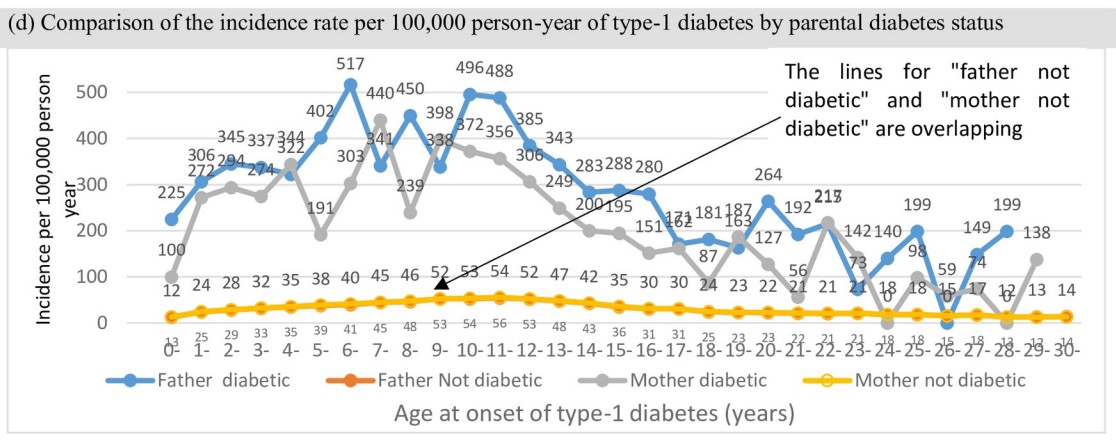

**Fig 1. Distribution of incidence of type-1 diabetes by selected children characteristics.**

associated with the birth cohort of the children. The younger the children the hazard of type-1 diabetes. For instance, those born in 1991–1995 had 18%, 1996–2000 (33%), 2001–2005 (38%), 2006–2010 (38%) and 2011–2015 (36%) higher hazard of type-1 diabetes than those in 1985–1990 birth cohort.

The hazard of type-1 diabetes was higher among children whose mothers do not smoke (aHR = 1.296; 95% CI: 1.240–1.354, p<0.001) than those whose mother smokes. The adjusted hazard of type-1 diabetes was 60% higher among children of both Swedish and other Nordic mothers than those of other European mothers while the hazard was about 76% higher among the children of both Swedish and other Nordic fathers than those of other European fathers. The hazard of type-1 diabetes increased with a lower level of paternal and maternal education. The highest hazard of type-1 diabetes was found among children whose parents were diabetic. Children of diabetic mothers and fathers were about 6 times (aHR = 6.419; 95% CI: 5.852–7.041, p<0.001) and 9 times (aHR = 8.808; 95% CI: 8.221–9.437, p<0.001) respectively more likely to be diabetic than those whose parents were not diabetic as shown in Table 4. However, maternal age and the parity of the index child were not significantly associated with the risk of type-1 diabetes. The visualization of the adjusted hazard ratios is presented in Fig 5.

## Discussions

This study was designed to compare the performance of different survival models in examining the risk of type-1 diabetes viz-a-viz mode of conception and assess if there are differentials in the risk of type-1 diabetes between children from fresh or frozen-thawed embryo transfers. In all, the models considered in this study are comparable as far as modelling the effect of conception modes on the risk of diabetes is concerned. While there were no significant differences in the risk of diabetes between children conceived spontaneously and by ART, the risk differed significantly among children from fresh and frozen-thawed embryo transfers. The measures of goodness of fit of models considered in this study were relatively lower in the flexible models. However, rather than likelihoods, the CPH model uses a different procedure and returns partial likelihood, which is also lower than the likelihood of the other models. Similarly, the AIC and the BIC which are measures of information lost by each of the models were more favourable to the flexible models. Similar approaches have been used to identify better models in the

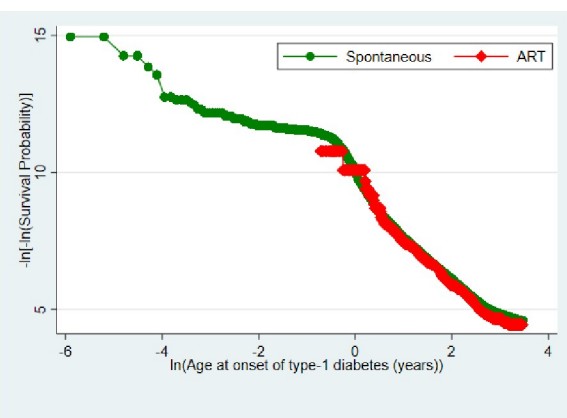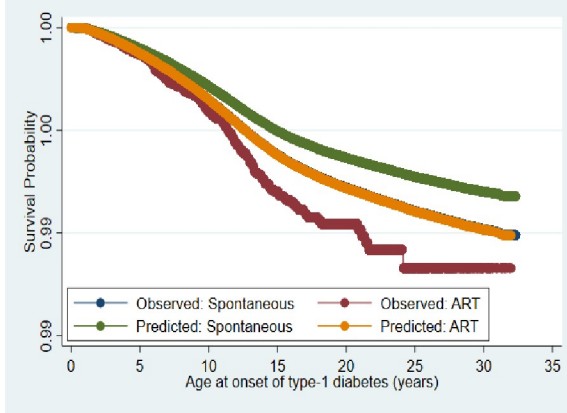

**Fig 2. Proportionality of survival curves of onset of type-1 diabetes among ART and spontaneously conceived children.**

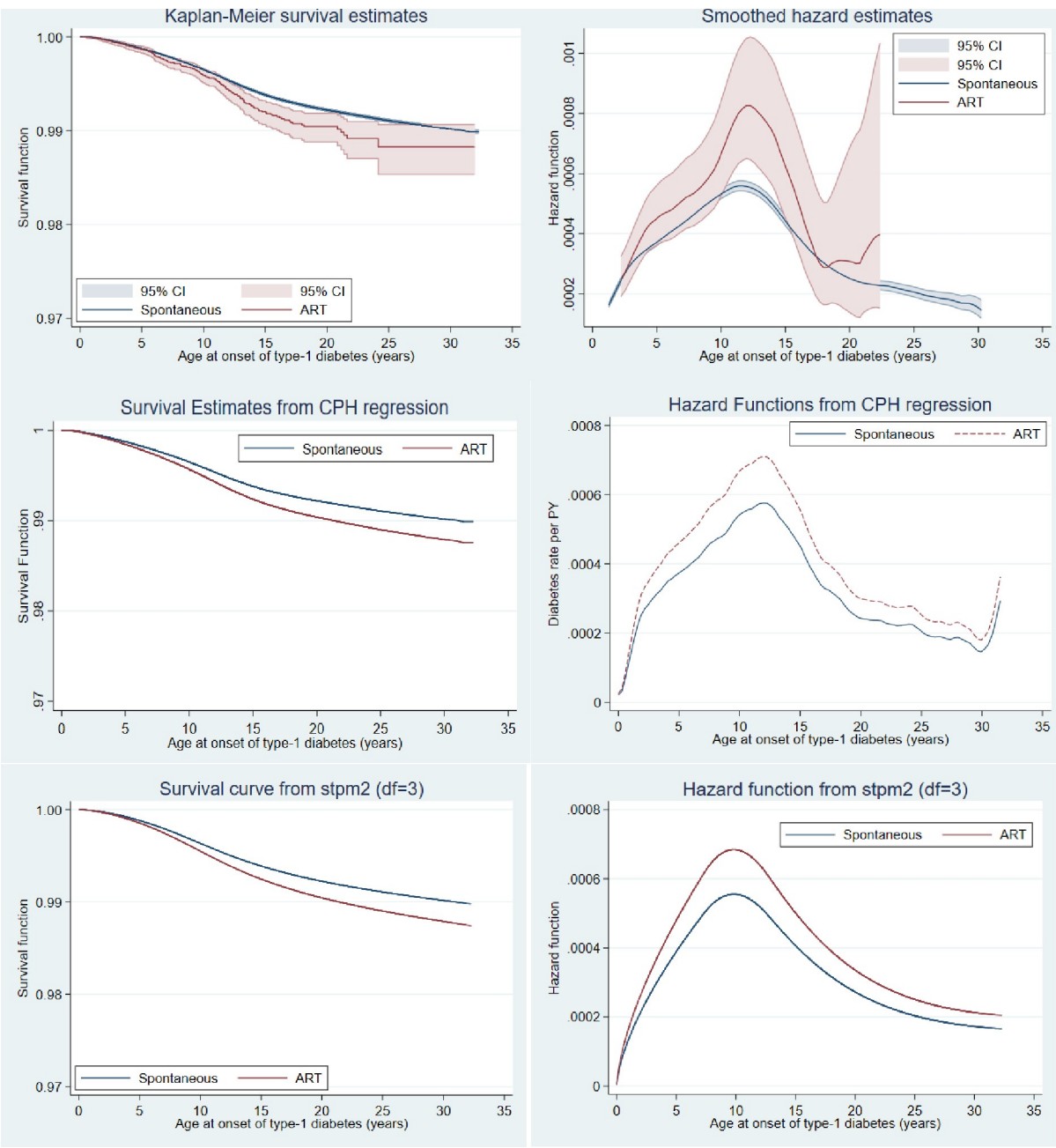

**Fig 3. Comparison of hazard and survival functions under different survival regression models.**

literature [31, 32]. Besides having the lowest information lost, the flexible model fitted the data most but this did not lead to large differences between the estimates from the models. At higher degrees of freedom, the flexible model fitted the data most and also showed that the occurrence of type-1 diabetes is not proportional to the baseline hazard but that the risk changed severally as the children grew older. We have applied the proportional hazard and other parametric, as well as the flexible parametric models to the data. A major strength of the flexible parametric model is its ability to account for non-proportional hazards, hence individual non-PH models were not explored in this study. However, the estimates from the flexible

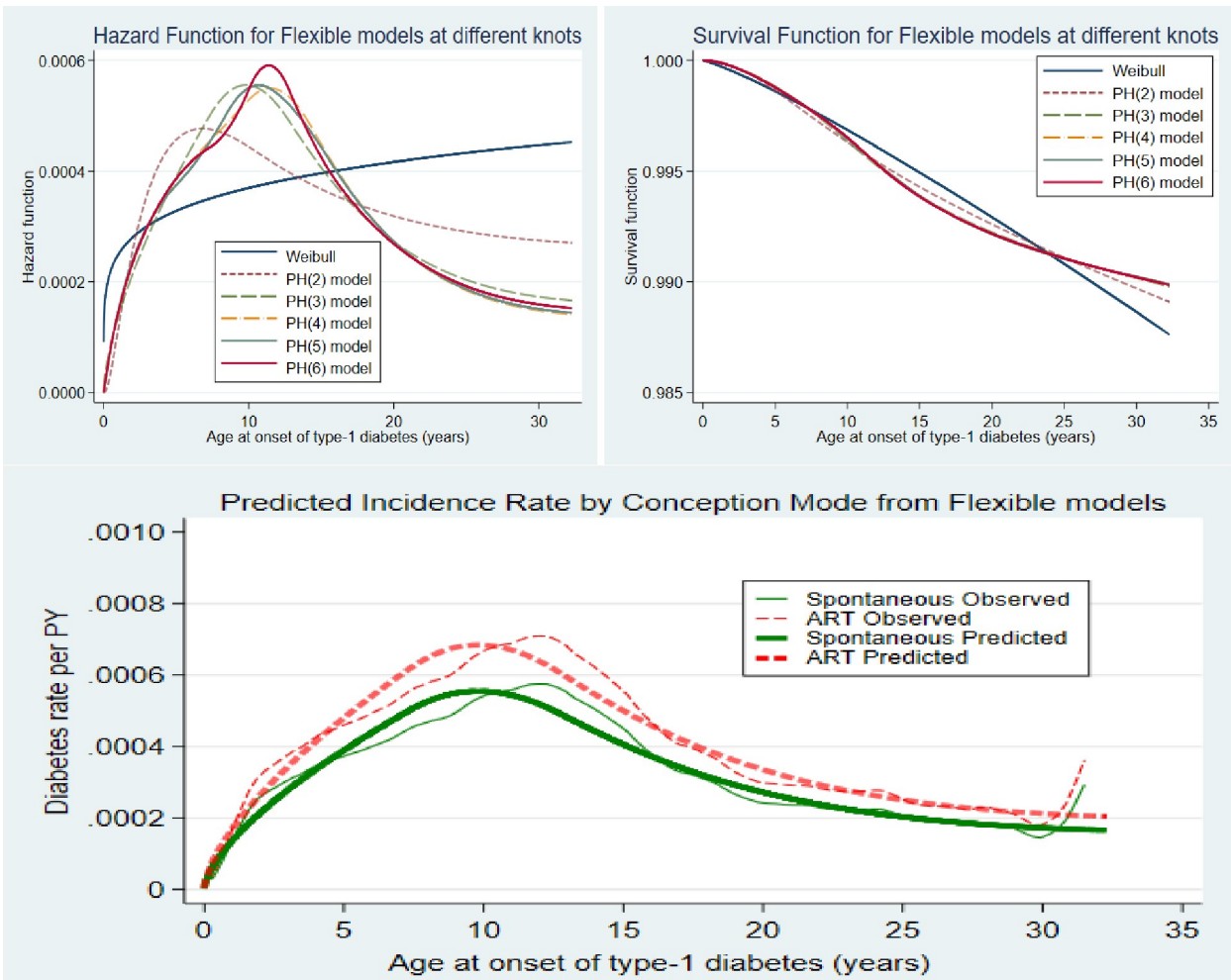

**Fig 4. Comparing the hazard and survival functions of the flexible models at different knots/splines.**

parametric and CPH models are similar when the proportional hazard assumptions were not violated.

The estimates from the adjusted hazard ratios from the CPH model was close to those obtained from the parametric PH and the flexible models. Lambert et al. had stated that rather than just a comparison of the AIC or BIC, it is a strength to draw conclusions from the estimates that do not change largely when different reasonable models are chosen [32]. Therefore, we can conclude that although the FPSR provided a relatively better model fit, the estimates from all the models considered in the current study are similar. With such, any of the models could be used. Among the FPSR models evaluated at different degrees of freedom, the loglikelihood, as well as the AIC and the BIC unanimously identified the 5-knot (6 degrees of freedom) flexible model as the best model. Our finding agreed with the recommendation of Royston et al., that flexible models with 5 or 6 degrees freedom will be more ideal for a sample size greater than 10,000 [19].

Nonetheless, the parameter estimates from the flexible and the CPH model are very close, despite the wide differences in all the measures of model fitness adopted in this study. These similarities underscore the strength of the CPH model when a "large" sample is followed up for a sufficiently "long" time [14]. There are possibilities that the proportional assumption

**Table 2. Comparison of the loglikelihood, AIC and BIC of the models considered.**

|  | Model | ll(null) | ll(model) | df | AIC | BIC |
|---|---|---|---|---|---|---|
| CPH | CPH | -260143.50* | -257910* | 27 | 515873.50 | 516222.80 |
| Parametric PH | Gompertz | -121461.50 | -118977 | 29 | 238012.30 | 238387.40 |
|  | Weibull | -121176.60 | -118547 | 29 | 237151.30 | 237526.40 |
|  | Exponential | -121462.00 | -118992 | 28 | 238039.70 | 238401.80 |
| Parametric AFT | Weibull | -121176.60 | -118547 | 29 | 237151.30 | 237526.40 |
|  | Exponential | -121462.00 | -118992 | 28 | 238039.70 | 238401.80 |
|  | Lognormal | -120706.30 | -118199 | 29 | 236455.40 | 236830.50 |
|  | Loglogistic | -121167.20 | -118534 | 29 | 237125.40 | 237500.50 |
| FPSR | df = 1 | -120890.80 | -118547 | 29 | 237151.30 | 237526.40 |
|  | df = 2 | -120890.80 | -117781 | 30 | 235621.20 | 236009.20 |
|  | df = 3 | -120890.80 | -117598 | 31 | 235258.00 | 235659.00 |
|  | df = 4 | -120890.80 | -117525 | 32 | 235114.20 | 235528.10 |
|  | df = 5 | -120890.80 | -117512 | 33 | 235090.60 | 235517.50 |
|  | df = 6 | -120890.80 | -117497 | 34 | 235062.50 | 235502.30 |

AIC Akaike Information Criteria BIC Bayesian Information Criteria df degrees of freedom ll loglikelihood PH Proportional Hazard AFT Accelerated Failure Rate

*partial likelihood

could be violated, especially in a long follow-up study of this nature. Bower et al affirmed the need to account for non-proportionality, if present, to get an insight into the natural history of the disease and biological process, and to make accurate predictions [14]. Rutherford et al. emphasized that the flexible models provide a more accurate estimate of complex hazard functions as well as unbiased estimates of hazard ratios than the parametric models [22]. The flexible model has a unique advantage of providing a direct estimate of relative and absolute effects besides the quantification of differences between survival and hazard functions using a time scale of interest [33–35].

However, literature has suggested that AFT models are better alternatives to the CPH in the analysis of time to event data, especially when there is no permanent effect in the context of the follow-up period [36–39]. The AFT models assume that the effect of a covariate is either to accelerate or decelerate the life course of an outcome, say diabetes type-1, by some constant. Whereas, the CPH model assumes that the hazard functions for any two patients with baseline $x$ vectors $x_1$ and $x_2$ are constrained to be proportional, upon which the estimation method is based [37–39]. Additionally, Lambert et al remarked that, unlike a CPH model, regression parameter estimates from AFT models are robust to neglected covariates and less affected by the chosen probability distribution [36].

Although the hazard of type-1 diabetes was higher among children conceived through ART than those conceived spontaneously in the bivariable analysis, we found no significant difference in the hazard of type-1 diabetes among the two groups of children in the adjusted models. This finding is corroborated in the literature by recent Danish studies [2, 3]. However, a further sub-group analysis controlling for other variables showed that the children conceived from frozen-thawed transfer had a significantly higher hazard of type-1 diabetes than those from fresh embryo transfer and children from spontaneous conception. This is in agreement with the previous study on the same cohort by Norrman et al. who reported a higher risk of type-1 diabetes among children from frozen-thawed embryo transfer than those from fresh embryo transfer (aHR: 1.52; 95% CI: 1.08–2.14) and those conceived spontaneously (aHR: 1.41; 95% CI: 1.05–1.89) [1]. This implies that the type of ART adopted is associated with the risk of type-1 diabetes.

**Table 3. Coefficients of the covariates from the fitted models.**

| Characteristics | CPH β(s.e) | Parametric PH (β(s.e)) Gompertz | Weibull | Exponential | Parametric AFT (β(s.e)) Weibull | Exponential | Lognormal | Loglogistic | Flexible Model (β(s.e)) df == 2 | df == 4 | df == 5 | df == 6 |
|---|---|---|---|---|---|---|---|---|---|---|---|---|
| ART | 0.07 (0.077) | 0.07 (0.077) | 0.06 (0.077) | 0.07(0.077) | 0.06 (0.054) | 0.06(0.067) | 0.05(0.057) | 0.06(0.054) | 0.07 (0.077) | 0.07 (0.077) | 0.07 (0.077) | 0.07 (0.078) |
| Sex | | | | | | | | | | | | |
| Male | 0.16 (0.018) | 0.16 (0.018) | 0.16 (0.018) | 0.16(0.018) | 0.13 (0.011) | 0.16(0.013) | 0.12(0.011) | 0.13(0.011) | 0.16 (0.018) | 0.16 (0.018) | 0.16 (0.018) | 0.17 (0.018) |
| Birth Cohort (1985–1990) | | | | | | | | | | | | |
| 1991–1995 | 0.16 (0.024) | 0.29 (0.028) | 0.24 (0.026) | 0.26(0.027) | 0.23 (0.013) | 0.24(0.016) | 0.23(0.014) | 0.23(0.013) | 0.21 (0.025) | 0.16 (0.024) | 0.16 (0.024) | 0.16 (0.024) |
| 1996–2000 | 0.26 (0.030) | 0.54 (0.039) | 0.43 (0.035) | 0.46(0.037) | 0.43 (0.012) | 0.43(0.015) | 0.41(0.013) | 0.43(0.012) | 0.37 (0.033) | 0.28 (0.031) | 0.28 (0.031) | 0.29 (0.031) |
| 2001–2005 | 0.26 (0.033) | 0.65 (0.048) | 0.46 (0.039) | 0.51(0.043) | 0.52 (0.012) | 0.46(0.015) | 0.48(0.013) | 0.52(0.012) | 0.39 (0.038) | 0.33 (0.035) | 0.32 (0.035) | 0.32 (0.035) |
| 2006–2010 | 0.21 (0.039) | 0.56 (0.055) | 0.26 (0.038) | 0.32(0.044) | 0.44 (0.015) | 0.26(0.023) | 0.39(0.016) | 0.44(0.015) | 0.28 (0.042) | 0.31 (0.043) | 0.32 (0.043) | 0.33 (0.044) |
| 2011–2015 | 0.08 (0.056) | 0.34 (0.073) | 0.15 (0.042) | 0.07(0.047) | 0.27 (0.031) | 0.15(0.056) | 0.28(0.028) | 0.27(0.031) | 0.31 (0.070) | 0.31 (0.070) | 0.31 (0.070) | 0.31 (0.070) |
| Maternal age(<20) | | | | | | | | | | | | |
| 20/24 | 0.07 (0.073) | 0.07 (0.073) | 0.07 (0.073) | 0.07(0.073) | 0.05 (0.051) | 0.07(0.063) | 0.06(0.053) | 0.05(0.051) | 0.07 (0.073) | 0.07 (0.073) | 0.07 (0.073) | 0.07 (0.073) |
| 25/29 | 0.09 (0.073) | 0.09 (0.073) | 0.09 (0.073) | 0.09(0.073) | 0.07 (0.050) | 0.09(0.061) | 0.08(0.052) | 0.07(0.050) | 0.09 (0.073) | 0.09 (0.073) | 0.09 (0.073) | 0.09 (0.073) |
| 30/39 | 0.10 (0.074) | 0.10 (0.074) | 0.10 (0.074) | 0.10(0.074) | 0.08 (0.050) | 0.10(0.061) | 0.09(0.051) | 0.08(0.050) | 0.10 (0.074) | 0.10 (0.074) | 0.10 (0.074) | 0.10 (0.074) |
| 40+ | 0.14 (0.091) | 0.14 (0.091) | 0.14 (0.091) | 0.14(0.091) | 0.11 (0.057) | 0.14(0.069) | 0.13(0.058) | 0.11(0.057) | 0.14 (0.091) | 0.14 (0.092) | 0.14 (0.091) | 0.14 (0.091) |
| Parity(> = 2) | | | | | | | | | | | | |
| First | 0.01 (0.016) | 0.01 (0.016) | 0.01 (0.016) | 0.01(0.016) | 0.01 (0.013) | 0.01(0.016) | 0.01(0.014) | 0.01(0.013) | 0.01 (0.016) | 0.01 (0.016) | 0.01 (0.016) | 0.01 (0.016) |
| Mother Smokes | | | | | | | | | | | | |
| No | 0.26 (0.029) | 0.26 (0.029) | 0.26 (0.029) | 0.26(0.029) | 0.21 (0.015) | 0.26(0.017) | 0.23(0.015) | 0.21(0.015) | 0.26 (0.029) | 0.26 (0.029) | 0.26 (0.029) | 0.26 (0.029) |
| Mother Country (Europa) | | | | | | | | | | | | |
| Sweden | 0.45 (0.092) | 0.45 (0.091) | 0.45 (0.092) | 0.45(0.092) | 0.36 (0.033) | 0.45(0.037) | 0.36(0.032) | 0.36(0.033) | 0.45 (0.091) | 0.45 (0.091) | 0.45 (0.091) | 0.45 (0.091) |
| Nordic | 0.47 (0.116) | 0.47 (0.116) | 0.46 (0.116) | 0.46(0.116) | 0.37 (0.040) | 0.46(0.046) | 0.38(0.040) | 0.37(0.040) | 0.47 (0.116) | 0.47 (0.116) | 0.47 (0.116) | 0.47 (0.116) |
| Other | 0.10 (0.067) | 0.10 (0.067) | 0.10 (0.067) | 0.10(0.067) | 0.08 (0.063) | 0.10(0.081) | 0.07(0.061) | 0.08(0.063) | 0.10 (0.067) | 0.10 (0.067) | 0.10 (0.067) | 0.10 (0.067) |
| Father Country (Europa) | | | | | | | | | | | | |
| Sweden | 0.57 (0.099) | 0.57 (0.099) | 0.57 (0.099) | 0.57(0.099) | 0.45 (0.028) | 0.57(0.032) | 0.45(0.028) | 0.45(0.028) | 0.57 (0.099) | 0.57 (0.099) | 0.57 (0.099) | 0.56 (0.100) |
| Nordic | 0.56 (0.127) | 0.56 (0.127) | 0.56 (0.127) | 0.56(0.127) | 0.45 (0.037) | 0.56(0.041) | 0.45(0.037) | 0.45(0.037) | 0.57 (0.127) | 0.57 (0.127) | 0.57 (0.127) | 0.57 (0.127) |
| Other | 0.32 (0.096) | 0.32 (0.096) | 0.32 (0.096) | 0.32(0.096) | 0.26 (0.043) | 0.32(0.051) | 0.26(0.042) | 0.26(0.043) | 0.32 (0.096) | 0.32 (0.096) | 0.32 (0.096) | 0.32 (0.096) |
| Mother education (>3 yr) | | | | | | | | | | | | |
| < = 9 years | 0.14 (0.036) | 0.13 (0.036) | 0.13 (0.036) | 0.13(0.036) | 0.11 (0.023) | 0.13(0.028) | 0.12(0.024) | 0.11(0.023) | 0.13 (0.036) | 0.14 (0.036) | 0.14 (0.036) | 0.14 (0.036) |

*(Continued)*

**Table 3.** (Continued)

| Characteristics | CPH β(s.e) | Parametric PH (β(s.e)) Gompertz | Weibull | Exponential | Parametric AFT (β(s.e)) Weibull | Exponential | Lognormal | Loglogistic | Flexible Model (β(s.e)) df = = 2 | df = = 4 | df = = 5 | df = = 6 |
|---|---|---|---|---|---|---|---|---|---|---|---|---|
| 10–12 years | 0.08 (0.022) | 0.08 (0.021) | 0.08 (0.022) | 0.08(0.022) | 0.06 (0.015) | 0.08(0.018) | 0.07(0.016) | 0.06(0.015) | 0.08 (0.022) | 0.08 (0.022) | 0.08 (0.022) | 0.08 (0.022) |
| Higher edu<3 years | 0.02 (0.025) | 0.02 (0.025) | 0.02 (0.025) | 0.02(0.025) | 0.02 (0.019) | 0.02(0.024) | 0.02(0.020) | 0.02(0.019) | 0.02 (0.025) | 0.02 (0.025) | 0.02 (0.025) | 0.02 (0.025) |
| Father education (>3 yr) | | | | | | | | | | | | |
| < = 9 years | 0.02 (0.030) | 0.02 (0.023) | 0.02 (0.023) | 0.02(0.023) | 0.02 (0.023) | 0.02(0.029) | 0.01(0.024) | 0.02(0.023) | 0.02 (0.030) | 0.02 (0.030) | 0.02 (0.030) | 0.02 (0.030) |
| 10–12 years | 0.09 (0.025) | 0.09 (0.025) | 0.09 (0.025) | 0.09(0.025) | 0.07 (0.017) | 0.09(0.021) | 0.07(0.018) | 0.07(0.017) | 0.09 (0.025) | 0.09 (0.025) | 0.09 (0.025) | 0.09 (0.025) |
| Higher edu<3 years | 0.04 (0.028) | 0.04 (0.028) | 0.05 (0.028) | 0.05(0.028) | 0.04 (0.021) | 0.05(0.026) | 0.04(0.022) | 0.04(0.021) | 0.04 (0.028) | 0.04 (0.028) | 0.04 (0.028) | 0.04 (0.028) |
| Mother Diabetic | 1.86 (0.302) | 1.86 (0.304) | 1.86 (0.302) | 1.86(0.302) | 1.48 (0.009) | 1.86(0.007) | 1.74(0.009) | 1.50(0.009) | 1.86 (0.303) | 1.86 (0.303) | 1.86 (0.303) | 1.86 (0.303) |
| Father Diabetic | 2.17 (0.309) | 2.18 (0.311) | 2.17 (0.309) | 2.17(0.309) | 1.73 (0.005) | 2.17(0.004) | 2.07(0.005) | 1.76(0.005) | 2.18 (0.310) | 2.18 (0.310) | 2.18 (0.310) | 2.19 (0.312) |

The current study established a higher risk of type-1 diabetes in the offspring of diabetic parents. Children of diabetic mothers were about 540% at higher risk of type-1 diabetes compared with children whose mothers are not diabetic. A similar pattern but a higher likelihood of risk (780%) was found among children whose fathers were diabetic. This finding aligns with the findings of previous studies [40, 41]. Diabetic parents pose a higher risk of type-1 diabetes to their children independent of the conception method. This is plausible as a systematic review has reported a 51% significant increase in diabetes mellitus among parents who conceived by ART compared with those who conceived spontaneously [42].

The sex of the children was significantly associated with the age at the onset of type-1 diabetes. The males were at higher risk of type-1 diabetes than females. Further research is necessary to understand the biological configuration of males that put them at higher risk of type-1 diabetes. Our study is however corroborated by earlier researches which showed that about three-fifths of children who developed type-1 diabetes were boys [43].

The median age at onset of type-1 diabetes among diabetic children was 10 years (Inter-quartile range: 6–14 years). There appears to be a generational shift in the age at the onset of type-1 diabetes among the children. The increase in the risk of type-1 diabetes in the studied population was linear with the lowest risk among the children born between 1985 and 1990 and highest (about 36% on average) among the birth cohorts of 2001–2005, 2006–2010, and 2011–2015. Our finding is at variance with the results of the Danish study which found no association between risk of type-1 diabetes and birth cohorts [3]. Although not reported in the results, our stratified birth cohort analysis showed that covariate effects remain constant over different cohorts.

We found a higher preponderance of type-1 diabetes among children whose mothers do not smoke compared with those whose mothers smoke. This finding is at variance with findings of a comparative study on the associations of parental smoking during pregnancy and childhood-onset of type-1 diabetes in the Norwegian Mother and Child Cohort Study and the Danish National Birth Cohort by Magnues et al. [44]. Magnus et al. reported that in both cohorts, maternal smoking beyond gestational week 12 was inversely associated with type-1 diabetes" and that "in the Norwegian register-based cohort, children of mothers who still smoked at the end of pregnancy had a lower risk of type-1 diabetes" [44]. The authors noted

**Table 4. Adjusted prognostic factors of type-1 diabetes among the studied Swedish cohort from CPH and FPSR model with 6 degrees of freedom.**

| Characteristics | FPSR Model (df = 6) | | CPH Model | |
|---|---|---|---|---|
| | aHR (95% CI) | p-value | aHR (95% CI) | p-value |
| Conception Mode | | | | |
| Spontaneous | 1.000 | | | |
| ART[a] | 1.070(0.929–1.232) | 0.346 | 1.068(0.927–1.230) | 0.361 |
| Spontaneous | 1.000 | | | |
| Fresh[b] | 1.010(0.862–1.184) | 0.902 | 1.008(0.860–1.182) | 0.917 |
| Frozen-thawed[b] | 1.368(1.013–1.847) | 0.041 | 1.361(1.011–1.834) | 0.044 |
| Sex | | | | |
| Female | 1.000 | | | |
| Male | 1.171(1.137–1.206) | 0.000 | 1.171(1.137–1.206) | 0.000 |
| Birth Cohort | | | | |
| 1985–1990 | 1.000 | | | |
| 1991–1995 | 1.179(1.132–1.228) | 0.000 | 1.168(1.121–1.217) | 0.000 |
| 1996–2000 | 1.332(1.273–1.395) | 0.000 | 1.297(1.240–1.358) | 0.000 |
| 2001–2005 | 1.380(1.313–1.451) | 0.000 | 1.297(1.234–1.363) | 0.000 |
| 2006–2010 | 1.382(1.299–1.470) | 0.000 | 1.234(1.160–1.311) | 0.000 |
| 2011–2015 | 1.359(1.228–1.504) | 0.000 | 1.085(0.981–1.199) | 0.114 |
| Maternal age (years) | | | | |
| <20 | 1.000 | | | |
| 20/24 | 1.073(0.940–1.226) | 0.296 | 1.074(0.940–1.226) | 0.295 |
| 25/29 | 1.093(0.958–1.246) | 0.186 | 1.093(0.958–1.246) | 0.185 |
| 30/39 | 1.104(0.967–1.260) | 0.143 | 1.104(0.967–1.260) | 0.142 |
| 40+ | 1.151(0.985–1.345) | 0.076 | 1.151(0.985–1.345) | 0.076 |
| Parity | | | | |
| > = 2 | 1.000 | | | |
| First | 1.007(0.975–1.040) | 0.677 | 1.007(0.975–1.039) | 0.688 |
| Smoking mother | | | | |
| Yes | | | | |
| No | 1.296(1.240–1.354) | 0.000 | 1.295(1.239–1.353) | 0.000 |
| Mother Country | | | | |
| Europa | 1.000 | | | |
| Sweden | 1.567(1.397–1.757) | 0.000 | 1.569(1.399–1.759) | 0.000 |
| Nordic | 1.596(1.384–1.841) | 0.000 | 1.594(1.382–1.839) | 0.000 |
| Other | 0.908(0.786–1.048) | 0.187 | 0.907(0.785–1.047) | 0.182 |
| Father Country | | | | |
| Europa | 1.000 | | | |
| Sweden | 1.767(1.583–1.972) | 0.000 | 1.769(1.585–1.975) | 0.000 |
| Nordic | 1.760(1.527–2.028) | 0.000 | 1.759(1.526–2.027) | 0.000 |
| Other | 1.380(1.204–1.582) | 0.000 | 1.378(1.203–1.580) | 0.000 |
| Mother Education | | | | |
| Higher edu> = 3 years | 1.000 | | | |
| < = 9 years | 1.145(1.076–1.219) | 0.000 | 1.145(1.076–1.219) | 0.000 |
| 10–12 years | 1.082(1.041–1.125) | 0.000 | 1.082(1.041–1.125) | 0.000 |
| Higher edu<3 years | 1.024(0.976–1.074) | 0.327 | 1.025(0.977–1.075) | 0.320 |
| Father Education | | | | |
| Higher edu> = 3 years | 1.000 | | | |
| < = 9 years | 1.022(0.965–1.082) | 0.458 | 1.023(0.966–1.084) | 0.431 |

(*Continued*)

**Table 4.** (Continued)

| Characteristics | FPSR Model (df = 6) | | CPH Model | |
|---|---|---|---|---|
| | aHR (95% CI) | p-value | aHR (95% CI) | p-value |
| 10–12 years | 1.094(1.046–1.144) | 0.000 | 1.095(1.047–1.145) | 0.000 |
| Higher edu<3 years | 1.045(0.991–1.102) | 0.105 | 1.046(0.992–1.103) | 0.099 |
| Mother Diabetic | 6.419(5.852–7.041) | 0.000 | 6.405(5.839–7.026) | 0.000 |
| Father Diabetic | 8.808(8.221–9.437) | 0.000 | 8.789(8.203–9.417) | 0.000 |

aHR adjusted Hazard Ratio, CPH Cox Proportional Hazard FSPR Flexible Parametric Survival Regression

_rcs Spline variables for the log baseline cumulative hazard Estimates from the FPSR model are significant but are not presented.

a,b fitted in separate models alongside the covariates

that their findings were in contrast to teratogens operating very early during development, but similar to what has been reported earlier for effects of maternal smoking on birth weight [45]. There could be pathways to the modification of the effect of smoking on diabetes. One such is the effect of low birth weight. Reduced risk of type-1 diabetes in children with low birth weight has been reported and low birth weight is more prevalent among smoking mothers [46].

The country of birth of the parents significantly predicted the risk of type-1 diabetes with higher risks among children of Swedish and other Nordic mothers and fathers compared with children whose either parent is born outside the Nordic countries. Existing literature had suggested that people from certain countries are more likely to be diabetic than others [1, 47–51]. Studies have shown a higher preponderance of type-1 diabetes in the Scandinavian countries [1, 49, 50]. Although the reasons for these differences are yet to be well articulated in the literature.

Educational attainment of parents was significantly associated with the risk of type-1 diabetes among the children. The risk of type-1 diabetes was higher among children of those who had a lower level of paternal and maternal education. Educational attainment could be associated with the amount of information at the disposal of parents which may or may not predispose a child to type-1 diabetes. Maternal age and the parity of the index child were not significant predictors of type-1 diabetes.

### Strength and limitations of the study

Our study has a major strength of a detailed analysis and comparison of the fit and performance of a different range of regression models including the semi-parametric proportional hazard, parametric proportional hazard and accelerated failure time models as well as the flexible model at different degrees of freedom to arrive at our conclusions. Also, we have used a large dataset that consists of over three million births spanning over 30 years of follow-up and nearly 19,000 cases from high validity registries. Hence, the estimated risks of type-1 diabetes were made with high statistical precision. Furthermore, our study is based on Swedish nationwide population-based registries, we, therefore, used data with almost complete coverage with no sampling or recall bias, leading to high generalizability, no loss to follow up, and almost complete ascertainment of type-1 diabetes cases. Besides, we adjusted for several potential confounders including the sex of the children, parental type-1 diabetes status, and country of birth of the parents. One of the limitations of our study was the limited time of exposure for those born after 2000. Some data were missing hence we may not have accounted for some confounders. The control group consisted of all non-ART singletons. A better control group might have been children to subfertile couples conceiving spontaneously. However, such a control group is almost impossible to identify.

## Hazard Ratios of Type−1 Diabetes

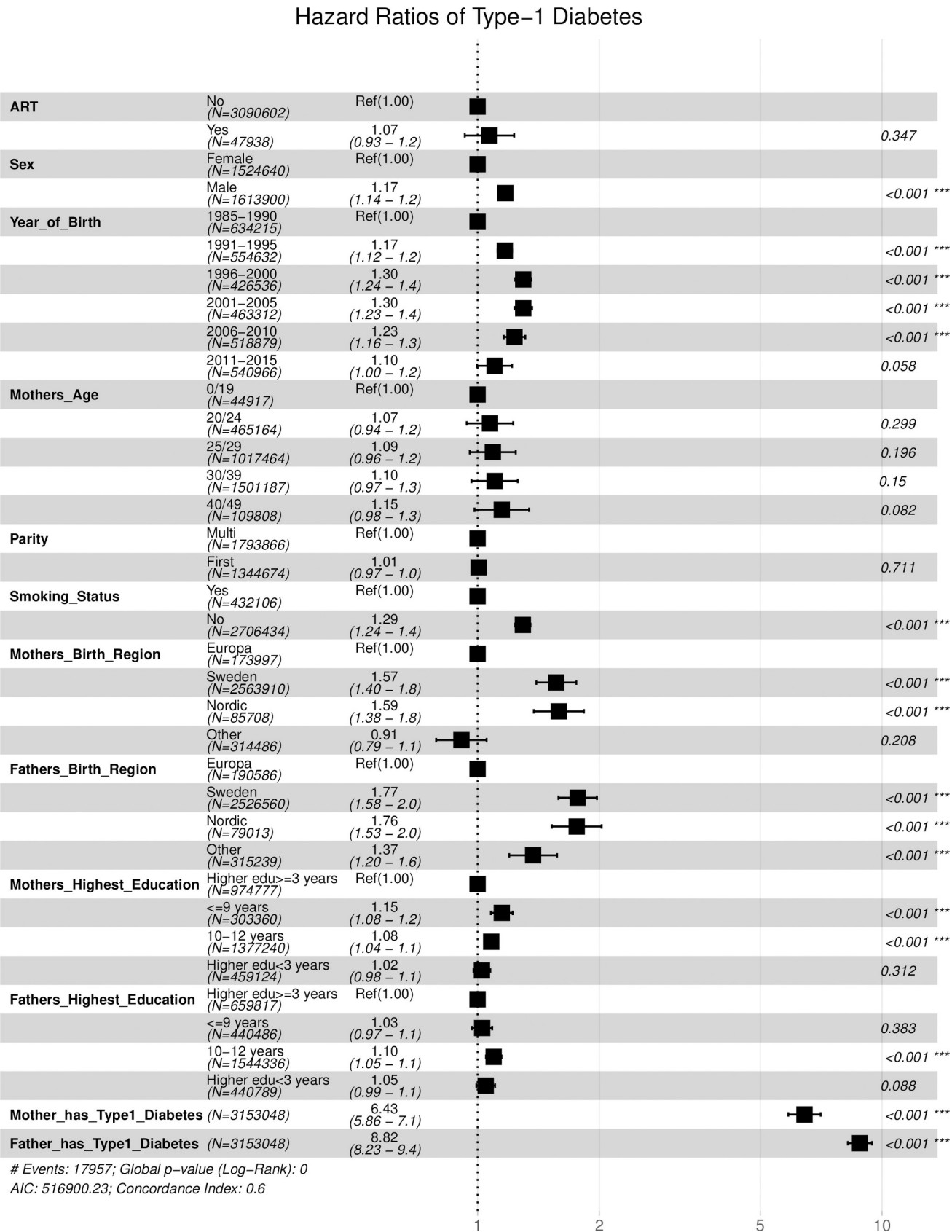

**Fig 5. The hazard ratios of the prognostic factors of type-1 diabetes among the studied Swedish cohort.**

## Conclusion

Our estimates from the different regression models including the semi-parametric proportional hazard, parametric proportional hazard models and the flexible model at different degrees of freedom were similar. This is an indication that the models performed similarly and any of them can be used to model the data. For the Swedish birth data used in the current study, we did not find any significant differences among the performances of the regression methods. There may be a need for a similar comparative analysis of Danish data used by Hargreave et al. and Kettner et al. [2, 3] so as to ascertain that the contradictions in their findings were not due to the methods of analysis. As shown in this study, for a very large sample followed up for a long time, the CPH model provided a comparative estimate with the FPSR model. Any of the models, especially the Cox semi-parametric model, parametric proportional hazard models and the flexible model at higher degrees of freedom are appropriate in modelling the risk of type-1 diabetes among ART and spontaneously-conceived children.

The results from our large population-based nationwide data were generally reliable, indicating that mode of conception could affect the risk of type-1 diabetes among the children especially when they are from frozen-thawed embryo transfer. Although we found no significant difference in the risk of type-1 diabetes among ART and spontaneously conceived children generally, our sub-analysis showed that children from frozen-thawed embryo transfer had a higher risk than other children. This is of some concern in view of the increasing number of frozen cycles all around the world. Estimates suggest that some 390,000 babies are now born each year globally through assisted fertility techniques including IVF and ICSI while the total number of ART children have exceeded eight million by 2019 [52, 53].

It's hard to generalise these findings to other settings. Although the considered models performed similarly in this specific setting, there's no guarantee that this will be the case with other data sources or diseases or other settings.

## Acknowledgments

The authors appreciate the logistic supports provided by the Consortium for Advanced Research and Training in Africa (CARTA) to AFF to visit the University of Gothenburg as part of his fellowship at the University of Warwick. The data analysis was done during the visit.

## Author Contributions

**Conceptualization:** Adeniyi Francis Fagbamigbe, Max Petzold.

**Data curation:** Adeniyi Francis Fagbamigbe, Emma Norrman, Christina Bergh.

**Formal analysis:** Adeniyi Francis Fagbamigbe.

**Investigation:** Christina Bergh, Ulla-Britt Wennerholm.

**Methodology:** Adeniyi Francis Fagbamigbe, Max Petzold.

**Supervision:** Emma Norrman, Ulla-Britt Wennerholm, Max Petzold.

**Visualization:** Adeniyi Francis Fagbamigbe.

**Writing – original draft:** Adeniyi Francis Fagbamigbe.

**Writing – review & editing:** Adeniyi Francis Fagbamigbe, Emma Norrman, Christina Bergh, Ulla-Britt Wennerholm, Max Petzold.

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
