## [Decision Letter · Decision Letter 0]

9 Feb 2021

PONE-D-21-00472

Conception modes and risk of Type-1 diabetes among 1985-2015 Swedish birth cohort: How robust are the survival analysis regression models?

PLOS ONE

Dear Dr. Fagbamigbe,

Thank you for submitting your manuscript to PLOS ONE. After careful consideration, we feel that it has merit but does not fully meet PLOS ONE’s publication criteria as it currently stands. Therefore, we invite you to submit a revised version of the manuscript that addresses the points raised during the review process.

Please clarify in the aims that if this manuscript is “a method paper with an application” or if it is “a medical paper with advanced methods”.

We look forward to receiving your revised manuscript.

Kind regards,

Yiqiang Zhan

Academic Editor

PLOS ONE

Journal Requirements:

2. In your ethics statement in the manuscript and in the online submission form, please ensure that you have discussed whether all data/samples were fully anonymized before you accessed them and/or whether the IRB or ethics committee waived the requirement for informed consent. If patients provided informed written consent to have data/samples from their medical records used in research, please include this information.

3. In the ethics statement in the manuscript and in the online submission form, please provide additional information about the patient records/samples used in your retrospective study, including the date range (month and year) during which patients' medical records/samples were accessed.

6. We note you have included a table to which you do not refer in the text of your manuscript. Please ensure that you refer to Table 3 in your text; if accepted, production will need this reference to link the reader to the Table.

Reviewers' comments:

Reviewer's Responses to Questions

**Comments to the Author**

1. Is the manuscript technically sound, and do the data support the conclusions?

Reviewer #1: Partly

Reviewer #2: Yes

Reviewer #3: Partly

Reviewer #4: Partly

2. Has the statistical analysis been performed appropriately and rigorously? 

Reviewer #1: Yes

Reviewer #2: Yes

Reviewer #3: No

Reviewer #4: No

3. Have the authors made all data underlying the findings in their manuscript fully available?

Reviewer #1: Yes

Reviewer #2: No

Reviewer #3: No

Reviewer #4: Yes

4. Is the manuscript presented in an intelligible fashion and written in standard English?

Reviewer #1: Yes

Reviewer #2: Yes

Reviewer #3: No

Reviewer #4: No

5. Review Comments to the Author

Reviewer #1: The authors of this study attempt to address two questions, 1) how does mode of conception affect the risk of type I diabetes in children, and 2) which survival model is best for the assessment of this association.

Main comments

The combination of the two above aims does not flow in my opinion, and is not adequately motivated. Who is the target audience of this paper? The medical research question of interest, seems to be relevant and interesting (although not my field of work), but the comparison of survival analysis models does not add to the current body of work in this area. Even if this did add to the question of “which is the best survival analysis model?” (which I’m not convinced has a definite answer), the comparison of methods here is undertaken in a situation where we do not know the underlying true answer to the question of interest. It is therefore, very difficult to determine which of the models is the best fitting. Furthermore, the selection of the best model according to the AIC and BIC is of course useful, but does not concretely determine which of the models fitted is the best model for the data. The AIC and BIC can indicate that different models are the better fitting model, and should be used as a guide for model selection along with knowledge about the specific subject area. In my opinion, the model selection process described in this paper, to some extent is what most researchers do in the background without presenting such results, but rather describe that they have chosen, say, a flexible parametric survival model based on the AIC and BIC.

For the above reasons, I would suggest that the authors of the paper focus on the research question of interest, i.e., the association between type I diabetes mode of conception, and refrain from presenting the comparison of survival analysis models in this paper.

Independent of the above comments, I think that some work could be done in the structuring of the paper to ensure a logical progression through the manuscript to help the reader’s understanding. The description of the survival analysis models in the introduction needs some more careful editing (see specific comments for examples), and I would suggest that detailed descriptions are instead presented in the methods section. There are results presented with no corresponding description in the methods section; the authors give detailed mathematical descriptions of the models they are going to fit, but no actual description of the models used for the analysis, i.e. how were certain variables modelled? Attributable fraction results are presented in the results with no description of what the measure is in the methods section. Another example is that the AIC and the BIC are described well, but the paper is lacking a sentence stating “We used these criteria to assess which of the above described models was the optimal model.” Additionally, some of the language used in the introduction is very non-specific; examples include the terms “adequate representation”, or “building on their strengths”. I would suggest the paper needs editing before it is at the standard for publication.

Specific comments

1. Ensure that initialisations are defined (e.g. SC, NPMLE)

2. Avoid the use contractions: “doesn’t” in the introduction

3. Add reference for Lebesgue measure

4. Some of the models are described as being proportional hazards models, and whilst the Cox model is most commonly called the Cox Proportional Hazards Model, the assumption of proportional hazards (PH) can be relaxed in these models and in the other models described.

5. Related, were non-proportional hazards models fitted? If not, why the focus on this in the description of the models? Perhaps the authors wanted to highlight an advantage of the flexible parametric survival models (that the PH assumption can be relaxed rather easily in comparison to the other survival models), but then this needs to be stated. If PHs are mentioned then a more thorough description would be a good idea to help the reader.

6. Line 118: Smooth hazard ratios should be smooth hazard rates.

7. Description of flexible parametric models (FPM) in the introduction: restricted cubic splines (RCS) do not need to model log time. In fact, when considering attained age as the time scale, restricted cubic splines are often used to model the untransformed time scale.

8. It is not clear to me why the comparison between the use of RCS in FPMs is compared to a linear function of log time in the introduction.

9. When presenting survival analyses, the timescale and event of interest are often presented alongside the exposure of interest and any confounding variables. I would suggest describing the analysis in this way instead of dependent/independent variables.

10. Related, a description to link the motivation behind the age distribution problems described in the introduction, and how this was dealt with in your analysis is needed (I think you did this by using age as the timescale?)

11. When incidence rates are so low, present them per 1,000 person-years (for example)

12. Was there missing data? If so, how much? How did you deal with this?

13. Figures need some cleaning up: ensure consistency between grayscale and colour figures, x-axes should be consistently named.

14. Figure 3: there is no description of the smoother used for the smoothed hazard estimates. “Stpm2” will not be clear for readers who are not familiar with FPM and/or Stata.

15. Figure 4, third panel: which degrees of freedom are presented?

16. I think the results you rely on are from the FPM with 6 degrees of freedom. This is quite a high number. Given the subject area, is the hazard function presented in figure 3 with 6 degrees of freedom reasonable to the authors?

17. Table 3: all the coefficient estimates need not be presented, only the exposure of interest and a description that the estimates were adjusted for the other variables. In particular, since the estimates for the restricted cubic spline coefficients are so often misinterpreted (and actually never presented in the context of answering a medical research question) I would suggest to remove these.

Reviewer #2: The advantages of this study are 1) compared statistical methods for exploring effects of different conception modes on a long-term outcome (Type 1 diabetes), using observational data from a nationwide register in Sweden with almost 31 years follow-up; 2) the authors are familiar with applications of parametric, flexible parametric and semi-parametric regression models to time-to-event data analysis. Sharing the codes would benefit the STATA users who want to do these similar comparisons.

Here are some comments for improvement:

1) In multivariable analysis, the adjustment was mad for child’s sex as well. Did the authors consider sex as a potential confounder or only an independent predictor for the outcome? It seems like sex of child was not associated with conception modes.

2) There is space to improve abstract: e.g. the second sentence repeated the information of the first sentence. As seen, the same cohort (data) had been applied in the Reference paper 1 for the same research question, both with hazard ratios as the association measure. Some conclusions were also similar. The authors could emphasize on the comparisons of methods and presenting corresponding results, presenting less similar conclusions as Ref 1 in this study.

3) Time-to-event was defined from baby born (i.e time zero) that has been accepted in literature. However, conception modes were determined at the beginning of pregnancy. There were a time window of gestational age that might differed between ART and spontaneous conception, due to ART being a risk factor of preterm birth. Then baby with ART would early expose to risk of type 1 diabetes (T1D). How the authors consider the following question: whether there is potential bias by gestational age?

4) Using singleton births from the same mothers, there might be more or less intra-correlation, especially for T1D showing familial aggregation, which could be discussed by the authors why didn’t take it account in this study.

5) Check the text carefully, e.g. minor typos: “Royston-Palmar” should be Royston-Parmar in several places.

Reviewer #3: The manuscript by Fagbamigbe et al. describes the results of an observational study on the association between conception mode and the risk of type-I diabetes using data from the Swedish birth register. Furthermore, they aim to compare the results of different survival models in their settings, with the hypothesis that using inappropriate models might explain contradictory findings that have been reported in the literature.

I think the applied analysis of the manuscript is interesting (nice work), but the more methodological parts need some extra work, as I feel the authors don’t fully accomplish what they aimed to do. In particular, I would like to see a more detailed comparison of the modelling approaches being studied, to show the effect of misspecifying a model or choosing a not-flexible-enough model for time to event data; statistical Monte Carlo simulation might even help selling the message as well.

More detailed comments (section by section) follow.

# Introduction

- Line 102/103: The Kaplan-Meier estimator can actually be used to investigate the effect of covariates, as you could in principle stratify the analysis (see e.g. the results of sts graph, by(covariate) in Stata) and still get distinct estimates of the survival function. Please reword that sentence. Having said that, of course this gets cumbersome when several covariates are to be investigated together, and therefore I agree that regression models are helpful in those settings.

- Line 119/122: I am not sure what the authors mean with the sentence “...and the fact that the cumulative hazard function or survival function may not be unbiased”. Please clarify.

# Flexible parametric survival regression model

- Please add a citation for software that is not part of Stata itself, e.g. the stpm2 Stata package used to fit the FPMs (https://www.stata-journal.com/article.html?article=st0165); it’s academic output by academic researchers, who need to seek funding to continue developing and supporting such packages.

# Results

- Some captions in Figure 1 are cropped away (e.g. the bottom panel, I feel like text ends abruptly), was there an issue with the online submission process or something like that? Also, several numbers overlap (e.g. panel b and d) making the graph really hard to read. Would it be possible to improve on that?

- Incidence rates per 1-person-year and per 100000-person-years are both presented, I think the former should be removed (as it is redundant and much harder to grasp compared to the same metric rescaled to 100000-person-years)

- How was the test for the equality of incidence rates conducted? Are the estimated measures fully unadjusted? It’s not clear from the text, please clarify.

- The log-log plot from figure 2 doesn’t show such a strong violation of the proportionality assumption, especially with such a large sample size; has that been tested further using other methods? How is the panel on the right testing the proportionality assumption?

- Are the hazard comparisons from fully adjusted models and non-parametric estimates? If so, please be careful that interpretation might not be the same and they might not be directly comparable.

- I wouldn’t say that the baseline hazard function looks more realistic with 6 df, as it is expected for it to be “more wobbly” with more degrees of freedom: please reword.

- When comparing the AIC/BIC/likelihood of the models, the Cox model returns a very different value: that is because the model uses (and returns) partial likelihood, therefore it is not possible to compare its value (nor AIC or BIC) to those of parametric and fully parametric model. This section needs to be re-written to fix this issue.

- In the same section, you see that AFT and PH models with the same baseline hazard distribution yield exactly the same fitted likelihood value: that is because they are equivalent, just with different parametrisations. This is not discussed at all in the manuscript, and it is particularly important as model coefficients from AFT and PH models have different interpretation.

- The authors are over-interpreting the significance tests for the coefficients of the spline for the baseline hazard function in FPMs, which don’t really have any meaningful interpretation (not directly, at least).

- The authors said they were comparing “robustness” of different methods, but such comparison is not present (they only compare a FPM with 6 df and a Cox model). I was expecting at least a comparison of the fitted coefficients from all models included in the comparison, to assess how estimates would change by choosing a not-flexible-enough model.

- When adjusting (and interpreting) for calendar year in the analysis, isn’t the interpretation of it just the effect of time (e.g. older cohorts have had more time to develop diabetes, and hence a higher risk)?

The higher risk for smoking mothers was surprising, as well as the difference with parents of non-nordic heritage. Any further insight on that?

- Can the observed effect of having diabetic parents be just genetics (and heritability of the trait)? I am not an expert on the topic, so it might be a silly comment (I apologise if so).

- The model estimates from FPM (6 df) and the CPH are very similar, but this is not news - there are several papers (e.g. the one by Rutherford et al cited by the authors) that show that model coefficients are insensitive to the choice of number of degrees of freedom (as long as enough flexibility is allowed). The Cox model fully solves the problem by not modelling the baseline hazard at all, so it’s not surprising that the two are so close to each other.

# Discussion

- The section on AIC/BIC needs to be re-written, the CPH with partial likelihood cannot be directly compared to the other methods that use full likelihood.

- The conclusion that “estimates from all models considered are similar” is not supported by data presented in the manuscipt, as I could only see estimates from FPM (6 df) and CPH.

- Line 388/389, model-based predictions are not showed in the manuscript, this sentence is not supported by data/plots in the ms.

- Line 399/400: it’s hard to say that ART -> type I diabetes without a study in a proper causal framework, I would suggest toning down that conclusion.

- Some comments on calendar time and genetics from the previous section apply here as well.

# Conclusion

- It is hard to conclude that the analysis method didn’t matter: (1) model estimates from other models are not showed, and (2) there might still be time-dependent effects or interactions that have not been studied thoroughly here. In fact, it would be nice to study time-dependent effects of e.g. treatment and so on (given the sample size), maybe as future research?

# Some typos and language

- Line 75, page 3, I would remove the *a* from the “Using a Danish data…” sentence;

- Line 90, page 4, I think it should be *subgroups* instead of *subgroup*;

- The word “determinate variable” is used throughout the ms, I find it a bit unusual even though I understand what the authors mean with that?

- I find the paper hard to read/follow at times (but it might just be me!), I wonder if the language could be simplified to improve readability of the ms?

Reviewer #4: General comment:

This paper investigates different statistical modelling approaches to assess if conception mode from ART influences risk of type 1 diabetes in children. Data comes from the Swedish Medical Birth Registry including births 1985-2015, an impressive thirty years study period. The paper has two aims, 1) to assess the medical research question regarding ART and type 1 diabetes and 2) to assess the methodological question by comparing different methods, which is a very nice combination and a good example of applied statistics research. However, this can be a challenge to combine, and the paper lacks some clarity and structure. I think the authors need to decide whether they are writing a medical paper with some advanced methods or if they are writing a methods paper with an application.

Main comments:

1. My main concern is that the authors have not utilized the strength of the FPSR, which is the straight-forward possibility to include non-proportional hazards. If indeed it was the intention to utilize the FPSR to explore interesting patterns of association for ART and type 1 diabetes in the data, it is unclear why this possibility was not pursued. As it is, the CPH and FPSR are nearly identical proportional hazards models and no benefit of the FPSR can be made, other than also obtaining absolute rates and survival measures directly from the model without additional post-estimation as for CPH. Fig 2 (right) also clearly shows non-proportional hazards (with very little effect of ART prior to age 12, and with stronger effect after age 12), yet the authors fit two proportional hazards models.

2. Methods, The data, row 219-225: The Data section needs to be extended, for example the data sources need to be explicitly spelled out, including what quality registers and register information from Statistics Sweden. What are the completeness of these data sources? Have the inclusion criteria changed over time? From which register was type 1 diabetes obtained, which diagnosis codes, etc. If the aim is to write a medical paper, then I think the data section should be first in the methods section, and then followed by the statistical methods section. If the aim is to write a methodological paper then the data section can be less.

3. Table 2: Why is the Log-likelihood so much higher for the CPH? Were all models fitted with the same covariates (i.e. same linear predictor of covariate effects), apart from the difference in baseline hazard parameterisation?

4. Table 3: The birth cohort effect does not make sense. The FPSR and CPH should yield similar results, if they are proportional hazards models with similar adjustment factors. Please clarify why the models give such different results.

Minor comments:

5. Abstract: In the results it is stated “hazard” of type 1 diabetes, but are the authors not estimating incidence rates of diabetes? Maybe clarify which disease measure is estimated rather than using the generic term hazard.

6. Introd: Can be shortened, and some text around previous literature can be moved to Discussion, I think.

7. Intro row 102: Explain term NPMLE.

8. Intro row 104: KM curves can be estimated by risk factor groups (covariates), but it is difficult to adjust for multiple confounders simultaneously (however, the curves can be standardized, as a form of adjustment). Standardization is mainly used for one to two variables at the time. Please clarify.

9. Does the methods section require all the formula given, or can it be simplified and referenced to original papers instead?

10. Methods: I don’t understand the “……………………………..(1)” notation in the formulas. Also notation “(i = 1, ………., N)” should perhaps be “(i=1,2,…,N)”. I don’t understand the notation “j= 1; : : : ;P;” or the notation “… … … … … .” or the notation “+ ⋯… … … …+”.

11. Methods row 187: “Odd scale” should be “odds scale” I think?

12. Methods row 210: The sentence “The position of the internal knots is usually in centiles computed as 100/df.” The placement of knots are at the centiles of event times, I believe.

13. Methods, Dependent variable, row 227: I don’t understand the sentence “The dependent variable is the censored timing (age) of the onset of type-1 diabetes among children. The time was censored on the date of the data collection, emigration and death.” In survival analysis the outcome is two-dimensional and defined by a survival time (with a start and an end), and an event indicator. Please clarify. What does “date of data collection” mean? Is this the date of extraction from the registers?

14. Results: How was the attributable fraction calculated, please explain in the methods section and give a reference.

15. Results, row 270-274: I don’t understand the added value of this section. Why is inference made on crude unadjusted rates? This is better done in adjusted models further down in the results section.

16. Results: Row 278: The log rank test is not a test of proportional hazards assumption, I think. Please clarify.

6. PLOS authors have the option to publish the peer review history of their article (what does this mean?). If published, this will include your full peer review and any attached files.

Reviewer #1: No

Reviewer #2: No

Reviewer #3: No

Reviewer #4: No

---

## [Author Response · Author response to Decision Letter 0]

21 Feb 2021

The Editor

PLOS ONE

Attn: Yiqiang Zhan

Academic Editor

PLOS ONE

PONE-D-21-00472

Conception modes and risk of Type-1 diabetes among 1985-2015 Swedish birth cohort: How robust are the survival analysis regression models?

Dear Dr. Fagbamigbe,

Thank you for submitting your manuscript to PLOS ONE. After careful consideration, we feel that it has merit but does not fully meet PLOS ONE’s publication criteria as it currently stands. Therefore, we invite you to submit a revised version of the manuscript that addresses the points raised during the review process.

Please clarify in the aims that if this manuscript is “a method paper with an application” or if it is “a medical paper with advanced methods”.

YES, IT IS A METHOD PAPER WITH AN APPLICATION. WE HAVE STATED THIS ACCORDINGLY

 WE HAVE COMPLIED

2. In your ethics statement in the manuscript and in the online submission form, please ensure that you have discussed whether all data/samples were fully anonymized before you accessed them and/or whether the IRB or ethics committee waived the requirement for informed consent. If patients provided informed written consent to have data/samples from their medical records used in research, please include this information.

WE HAVE COMPLIED 3. In the ethics statement in the manuscript and in the online submission form, please provide additional information about the patient records/samples used in your retrospective study, including the date range (month and year) during which patients' medical records/samples were accessed.

 WE HAVE COMPLIED

THE SWEDISH NATIONAL BOARD OF HEALTH AND WELFARE, AND STATISTICS SWEDEN (SCB) PLACED ETHICAL RESTRICTIONS ON SHARING THE DATA PUBLICLY. 

THE SWEDISH NATIONAL BOARD OF HEALTH AND WELFARE, AND STATISTICS SWEDEN (SCB) PLACED ETHICAL RESTRICTIONS ON SHARING THE DATA PUBLICLY. THE RESTRICTION WAS CONCURRED TO BY THE REGIONAL ETHICAL COMMITTEE AT THE UNIVERSITY OF GOTHENBURG, SWEDEN.

THERE ARE RESTRICTIONS. SO WE CAN’T UPLOAD ANY PART OF THE DATA

WE HAVE COMPLIED

6. We note you have included a table to which you do not refer in the text of your manuscript. Please ensure that you refer to Table 3 in your text; if accepted, production will need this reference to link the reader to the Table.

WE HAVE COMPLIED 

Reviewers' comments:

Reviewer's Responses to Questions

Comments to the Author

Reviewer #1: The authors of this study attempt to address two questions, 1) how does mode of conception affect the risk of type I diabetes in children, and 2) which survival model is best for the assessment of this association.

Main comments

The combination of the two above aims does not flow in my opinion, and is not adequately motivated. Who is the target audience of this paper? The medical research question of interest, seems to be relevant and interesting (although not my field of work), but the comparison of survival analysis models does not add to the current body of work in this area. Even if this did add to the question of “which is the best survival analysis model?” (which I’m not convinced has a definite answer), the comparison of methods here is undertaken in a situation where we do not know the underlying true answer to the question of interest. It is therefore, very difficult to determine which of the models is the best fitting. Furthermore, the selection of the best model according to the AIC and BIC is of course useful, but does not concretely determine which of the models fitted is the best model for the data. The AIC and BIC can indicate that different models are the better fitting model, and should be used as a guide for model selection along with knowledge about the specific subject area. In my opinion, the model selection process described in this paper, to some extent is what most researchers do in the background without presenting such results, but rather describe that they have chosen, say, a flexible parametric survival model based on the AIC and BIC.

For the above reasons, I would suggest that the authors of the paper focus on the research question of interest, i.e., the association between type I diabetes mode of conception, and refrain from presenting the comparison of survival analysis models in this paper.

THANK YOU. WE AGREE WITH OUR REVIEWERS COMMENT, BUT THE AIM IS TO BE ABLE TO EVALUATE THE MODEL THAT FIT THE DATA TO AN EXTENT. OUR WORK IS A METHOD PAPER WITH AN APPLICATION. WE TOTALLY UNDERSTAND AND AGREE THAT THAT THERE MAY NOT BE THE “ULTIMATE BEST” MODEL BECAUSE IT IS IMPRACTICABLE TO CONSIDER ALL MODELS. HOWEVER, WE FEEL THAT THE MODELS SHOULD BE RETAINED, ESPECIALLY AS THE OTHER REVIEWERS ARE COMFORTABLE WITH IT.

Independent of the above comments, I think that some work could be done in the structuring of the paper to ensure a logical progression through the manuscript to help the reader’s understanding. The description of the survival analysis models in the introduction needs some more careful editing (see specific comments for examples), and I would suggest that detailed descriptions are instead presented in the methods section. There are results presented with no corresponding description in the methods section; the authors give detailed mathematical descriptions of the models they are going to fit, but no actual description of the models used for the analysis, i.e. how were certain variables modelled? Attributable fraction results are presented in the results with no description of what the measure is in the methods section. Another example is that the AIC and the BIC are described well, but the paper is lacking a sentence stating “We used these criteria to assess which of the above described models was the optimal model.” Additionally, some of the language used in the introduction is very non-specific; examples include the terms “adequate representation”, or “building on their strengths”. I would suggest the paper needs editing before it is at the standard for publication.

WE APPRECIATE THESE CRITICAL AND CONSTRUCTIVE SUGGESTIONS. WE HAVE TAKEN ALL THE SUGGESTIONS

Specific comments

1. Ensure that initialisations are defined (e.g. SC, NPMLE): 

THANK YOU, THIS HAS BEEN ADDED

2. Avoid the use contractions: “doesn’t” in the introduction: 

THANK YOU, THIS HAS BEEN ADDED

3. Add reference for Lebesgue measure; 

THIS HAS BEEN ADDED

4. Some of the models are described as being proportional hazards models, and whilst the Cox model is most commonly called the Cox Proportional Hazards Model, the assumption of proportional hazards (PH) can be relaxed in these models and in the other models described. 

WE AGREE. THANK YOU. IT HAS BEEN ADDED

5. Related, were non-proportional hazards models fitted? If not, why the focus on this in the description of the models? Perhaps the authors wanted to highlight an advantage of the flexible parametric survival models (that the PH assumption can be relaxed rather easily in comparison to the other survival models), but then this needs to be stated. If PHs are mentioned then a more thorough description would be a good idea to help the reader.

THANK YOU. WE HAVE REVIEWED THIS

6. Line 118: Smooth hazard ratios should be smooth hazard rates.

THANK YOU. IT HAS BEEN CORRECTED

7. Description of flexible parametric models (FPM) in the introduction: restricted cubic splines (RCS) do not need to model log time. In fact, when considering attained age as the time scale, restricted cubic splines are often used to model the untransformed time scale.

THANK YOU. IT HAS BEEN CORRECTED

8. It is not clear to me why the comparison between the use of RCS in FPMs is compared to a linear function of log time in the introduction.

THE UNNECESSARY PHRASE HAS BEEN EXPUNGED

9. When presenting survival analyses, the timescale and event of interest are often presented alongside the exposure of interest and any confounding variables. I would suggest describing the analysis in this way instead of dependent/independent variables.

THANK YOU. IT HAS BEEN CORRECTED IN THE MANUSCRIPT

10. Related, a description to link the motivation behind the age distribution problems described in the introduction, and how this was dealt with in your analysis is needed (I think you did this by using age as the timescale?) 

YES THE CHILDREN AGE WERE USED AS THE TIME SCALE

11. When incidence rates are so low, present them per 1,000 person-years (for example) 

WE RECOGNISED NTHIS AND WE ALREADY PROVIDED THE RATES PER 100,000 PERSON-YEAR IN THE LAST COLUMN OF TABLE 1

12. Was there missing data? If so, how much? How did you deal with this?

THERE ARE NO MISSING DATA

13. Figures need some cleaning up: ensure consistency between grayscale and colour figures, x-axes should be consistently named. 

THANK YOU. WE HAVE REPRODUCED THE FIGURES AND HARMONIZED THE LABELS

14. Figure 3: there is no description of the smoother used for the smoothed hazard estimates. “Stpm2” will not be clear for readers who are not familiar with FPM and/or Stata.

THANK YOU, WE HAVE PROVIDED ADDITIONAL INFORMATION ON STPM2

15. Figure 4, third panel: which degrees of freedom are presented?

THIS ARE THE DEGREES OF FREEDOM FOR THE DIFFERENIT KNOTS IN THE FLEXIBLE MODEL

16. I think the results you rely on are from the FPM with 6 degrees of freedom. This is quite a high number. Given the subject area, is the hazard function presented in figure 3 with 6 degrees of freedom reasonable to the authors?

YES, WE DIDN’T PRODUCE ANY GRAPH IN FIGURE 3 WITH 6 DEGREES OF FREEDOM, RATHER WITH 3 DEGREES OF FREEDOM. THE FIGURE 3 SHOWED EXPLORATORY ON THE AVERAGE LEVEL. THE COMPARISON OF THE DEGREES OF FREEDOM WERE SHOWED IN FIGURE4 WHERE 6 DEGREES FREEDOM WAS FOUND TO BE BETTER THAN OTHERS. 

17. Table 3: all the coefficient estimates need not be presented, only the exposure of interest and a description that the estimates were adjusted for the other variables. In particular, since the estimates for the restricted cubic spline coefficients are so often misinterpreted (and actually never presented in the context of answering a medical research question) I would suggest to remove these.

THANK YOU. WE HAVE REMOVED THE RESTRICTED CUBIC SPLINE COEFFICIENTS TO AVOID MISINTERPRETED BUT RETAINED THE OTHER COVARIATES BECAUSE THERE ARE NECESSARY TO ENSURE THAT READERS UNDERSTAND THE EFFECTS OF THESSE COVARIATES

Reviewer #2: The advantages of this study are 1) compared statistical methods for exploring effects of different conception modes on a long-term outcome (Type 1 diabetes), using observational data from a nationwide register in Sweden with almost 31 years follow-up; 2) the authors are familiar with applications of parametric, flexible parametric and semi-parametric regression models to time-to-event data analysis. Sharing the codes would benefit the STATA users who want to do these similar comparisons.

THANK YOU. WE CAN PROVIDE THE CODES

Here are some comments for improvement:

1) In multivariable analysis, the adjustment was mad for child’s sex as well. Did the authors consider sex as a potential confounder or only an independent predictor for the outcome? It seems like sex of child was not associated with conception modes.

NO, WE CONSIDERED SEX. AS SHOWN IN TABLE 2, THE RISK OF dm WAS 17% HIGHER AMONG THE MALES THAN THE FEMALES

2) There is space to improve abstract: e.g. the second sentence repeated the information of the first sentence. As seen, the same cohort (data) had been applied in the Reference paper 1 for the same research question, both with hazard ratios as the association measure. Some conclusions were also similar. The authors could emphasize on the comparisons of methods and presenting corresponding results, presenting less similar conclusions as Ref 1 in this study.

THANK YOU. IT HAS BEEN CORRECTED

3) Time-to-event was defined from baby born (i.e time zero) that has been accepted in literature. However, conception modes were determined at the beginning of pregnancy. There were a time window of gestational age that might differed between ART and spontaneous conception, due to ART being a risk factor of preterm birth. Then baby with ART would early expose to risk of type 1 diabetes (T1D). How the authors consider the following question: whether there is potential bias by gestational age?

NO, THE DATA OWNERS DID NOT CAPTURE THIS IMPORTANT INFORMATION.

4) Using singleton births from the same mothers, there might be more or less intra-correlation, especially for T1D showing familial aggregation, which could be discussed by the authors why didn’t take it account in this study.

AGAIN, THIS INFORMATION WAS NOT PROVIDED IN THE DATA REGISTER

5) Check the text carefully, e.g. minor typos: “Royston-Palmar” should be Royston-Parmar in several places.

THANK YOU. THIS AND ITS OTHER INSTANCES, HAVE BEEN CORRECTED

Reviewer #3: The manuscript by Fagbamigbe et al. describes the results of an observational study on the association between conception mode and the risk of type-I diabetes using data from the Swedish birth register. Furthermore, they aim to compare the results of different survival models in their settings, with the hypothesis that using inappropriate models might explain contradictory findings that have been reported in the literature.

I think the applied analysis of the manuscript is interesting (nice work), but the more methodological parts need some extra work, as I feel the authors don’t fully accomplish what they aimed to do. In particular, I would like to see a more detailed comparison of the modelling approaches being studied, to show the effect of misspecifying a model or choosing a not-flexible-enough model for time to event data; statistical Monte Carlo simulation might even help selling the message as well.

More detailed comments (section by section) follow.

# Introduction

- Line 102/103: The Kaplan-Meier estimator can actually be used to investigate the effect of covariates, as you could in principle stratify the analysis (see e.g. the results of sts graph, by(covariate) in Stata) and still get distinct estimates of the survival function. Please reword that sentence. Having said that, of course this gets cumbersome when several covariates are to be investigated together, and therefore I agree that regression models are helpful in those settings.

THANK YOU. THIS HAS BEEN CORRECTED

- Line 119/122: I am not sure what the authors mean with the sentence “...and the fact that the cumulative hazard function or survival function may not be unbiased”. Please clarify.

THANK YOU. THIS HAS BEEN CLARIFIED

# Flexible parametric survival regression model

- Please add a citation for software that is not part of Stata itself, e.g. the stpm2 Stata package used to fit the FPMs (https://www.stata-journal.com/article.html?article=st0165); it’s academic output by academic researchers, who need to seek funding to continue developing and supporting such packages.

THANK YOU. WE AGREE, IT WAS PORIGINALLY CITED AS REF 18, WE HAVE NOW INSERTED STPM2 AND CITED IT ACCORDINGLY AS REF 19.

# Results

- Some captions in Figure 1 are cropped away (e.g. the bottom panel, I feel like text ends abruptly), was there an issue with the online submission process or something like that? Also, several numbers overlap (e.g. panel b and d) making the graph really hard to read. Would it be possible to improve on that?

THANK YOU, WE HAVE CORRECTED THE GRAPHS

- Incidence rates per 1-person-year and per 100000-person-years are both presented, I think the former should be removed (as it is redundant and much harder to grasp compared to the same metric rescaled to 100000-person-years)

THANK YOU, WE FELT WE SHOULDN’T RETAIN ONLY ONE BUT THE FORMER IS NECESSARY TO UNDERSTAND AND COMPARE WITH MOST OF THE HAZARD GRAPHS WHILE THE LATER IS NECESSARY FOR COMPREHENSION

- How was the test for the equality of incidence rates conducted? Are the estimated measures fully unadjusted? It’s not clear from the text, please clarify.

THANK YOU, THE ESTIMATED MEASURES FOR EQUALITY OF INCIDENCE RATES WERE NOT ADJUSTED FOR COVARIATES

- The log-log plot from figure 2 doesn’t show such a strong violation of the proportionality assumption, especially with such a large sample size; has that been tested further using other methods? How is the panel on the right testing the proportionality assumption?

WE AGREE, IT DOESN’T SHOW A STRONG VIOLATION BUT ITS PRESENCE. THE RIGHT PANEL SHOWS AN ALTERNATIVE TEST OF NON-PROPORTIONALITY OF KAPLAN MEIER SURVIVAL CURVES OF THE OBSERVED AND PREDICTED ART ESTIMATES AT VARIOUS AGES OF THE CHILDREN

- Are the hazard comparisons from fully adjusted models and non-parametric estimates? If so, please be careful that interpretation might not be the same and they might not be directly comparable. 

- I wouldn’t say that the baseline hazard function looks more realistic with 6 df, as it is expected for it to be “more wobbly” with more degrees of freedom: please reword.

- When comparing the AIC/BIC/likelihood of the models, the Cox model returns a very different value: that is because the model uses (and returns) partial likelihood, therefore it is not possible to compare its value (nor AIC or BIC) to those of parametric and fully parametric model. This section needs to be re-written to fix this issue.

THE HAZARD COMPARISONS ARE FROM UNADJUSTED MODELS AND NON-PARAMETRIC ESTIMATES. WE HAVE REWORDED THE STATEMENT ON THE COMPARISON OF THE DIFFERENT HAZARD FUNCTIONS AT DIFFERENT DEGREES OF FREEDOM. WE HAVE ALSO IMPROVED ON THE STATEMENT ON THE CPH LIKELIHOOD

- In the same section, you see that AFT and PH models with the same baseline hazard distribution yield exactly the same fitted likelihood value: that is because they are equivalent, just with different parametrisations. This is not discussed at all in the manuscript, and it is particularly important as model coefficients from AFT and PH models have different interpretation.

- The authors are over-interpreting the significance tests for the coefficients of the spline for the baseline hazard function in FPMs, which don’t really have any meaningful interpretation (not directly, at least).

WE HAVE EXPLAINED THE SIMILARITY IN THE LIKELIHOOD OF THE AFT AND PH MODLES AND ALSO REMOVED UNNECESSARY INTERPRETATION OF THE SPLINE

- The authors said they were comparing “robustness” of different methods, but such comparison is not present (they only compare a FPM with 6 df and a Cox model). I was expecting at least a comparison of the fitted coefficients from all models included in the comparison, to assess how estimates would change by choosing a not-flexible-enough model.

OUR COMPARISON WERE STEP-WISE, WE FIRST USED THE AIC AND BIC TO SELECT MODELS AND THEN PRESENTED THE ESTIMATES OF THE SELECTED MODELS

- When adjusting (and interpreting) for calendar year in the analysis, isn’t the interpretation of it just the effect of time (e.g. older cohorts have had more time to develop diabetes, and hence a higher risk)?

The higher risk for smoking mothers was surprising, as well as the difference with parents of non-nordic heritage. Any further insight on that?

- Can the observed effect of having diabetic parents be just genetics (and heritability of the trait)? I am not an expert on the topic, so it might be a silly comment (I apologise if so).

YES, WE USED THE CHILDREN AGE AS THE TIMESCALE TO CONTROL FOR THE DIFFERENCES IN EXPOSURE. THERE ARE INSIGHTS INTO WHY CHILDREN OF SMOKING MOTHERS AS WELL AS PARENTS OF NON-NORDIC HERITAGE HAD HIGHER RISK OF DM. YES, SOME LITERATURE HAVE ESTABLISHED HERITABILITY OF THE DM TRAIT (SPURR ET AL AND WHINCUP ET AL).

- The model estimates from FPM (6 df) and the CPH are very similar, but this is not news - there are several papers (e.g. the one by Rutherford et al cited by the authors) that show that model coefficients are insensitive to the choice of number of degrees of freedom (as long as enough flexibility is allowed). The Cox model fully solves the problem by not modelling the baseline hazard at all, so it’s not surprising that the two are so close to each other.

THANK YOU

# Discussion

- The section on AIC/BIC needs to be re-written, the CPH with partial likelihood cannot be directly compared to the other methods that use full likelihood.

- The conclusion that “estimates from all models considered are similar” is not supported by data presented in the manuscipt, as I could only see estimates from FPM (6 df) and CPH.

- Line 388/389, model-based predictions are not showed in the manuscript, this sentence is not supported by data/plots in the ms.

- Line 399/400: it’s hard to say that ART -> type I diabetes without a study in a proper causal framework, I would suggest toning down that conclusion.

- Some comments on calendar time and genetics from the previous section apply here as well.

WE AGREE WITH ALL THE COMMENTS AND HAVE REPHRASED THE STATEMENT ON AIC/BIC. THE CONCLUSION OF EQUALITY OF ESTIMATES FROM THE DIFFERENT MODELS HAVE BEEN EXPUNGED. THE STATEMENT ON MODEL-BASED PREDICTIONS HAVE BEEN REVISED. THE WRONG STATEMENT HAS BEEN REMOVED. THE WORD “AFFECT” HAS BEEN MODIFIED TO READ “ASSOCIATED”. 

# Conclusion

- It is hard to conclude that the analysis method didn’t matter: (1) model estimates from other models are not showed, and (2) there might still be time-dependent effects or interactions that have not been studied thoroughly here. In fact, it would be nice to study time-dependent effects of e.g. treatment and so on (given the sample size), maybe as future research?

THANK YOU. WE TOTALLY AGREE THAT THE STUDY OF TIME-DEPENDENT EFFECTS SUCH AS TREATMENT AND SO ON ARE WORTHY OF FUTURE RESEARCH PROVIDED THE DATA COULD BE MADE AVAILABLE

# Some typos and language

- Line 75, page 3, I would remove the *a* from the “Using a Danish data…” sentence;

- Line 90, page 4, I think it should be *subgroups* instead of *subgroup*;

- The word “determinate variable” is used throughout the ms, I find it a bit unusual even though I understand what the authors mean with that?

- I find the paper hard to read/follow at times (but it might just be me!), I wonder if the language could be simplified to improve readability of the ms?

THANK YOU, WE HAVE CORRECTED THE HIGHLIGHTED TYPOS AND CARRIED OUT A LANGUAGE EDIT

Reviewer #4: General comment:

This paper investigates different statistical modelling approaches to assess if conception mode from ART influences risk of type 1 diabetes in children. Data comes from the Swedish Medical Birth Registry including births 1985-2015, an impressive thirty years study period. The paper has two aims, 1) to assess the medical research question regarding ART and type 1 diabetes and 2) to assess the methodological question by comparing different methods, which is a very nice combination and a good example of applied statistics research. However, this can be a challenge to combine, and the paper lacks some clarity and structure. I think the authors need to decide whether they are writing a medical paper with some advanced methods or if they are writing a methods paper with an application.

Main comments:

1. My main concern is that the authors have not utilized the strength of the FPSR, which is the straight-forward possibility to include non-proportional hazards. If indeed it was the intention to utilize the FPSR to explore interesting patterns of association for ART and type 1 diabetes in the data, it is unclear why this possibility was not pursued. As it is, the CPH and FPSR are nearly identical proportional hazards models and no benefit of the FPSR can be made, other than also obtaining absolute rates and survival measures directly from the model without additional post-estimation as for CPH. Fig 2 (right) also clearly shows non-proportional hazards (with very little effect of ART prior to age 12, and with stronger effect after age 12), yet the authors fit two proportional hazards models.

WE DID PURSUED THE USE OF FPSR TO EXPLORE INTERESTING PATTERNS OF ASSOCIATION FOR ART AND TYPE 1 DIABETES IN THE DATA AND OUR CONCLUSION.

2. Methods, The data, row 219-225: The Data section needs to be extended, for example the data sources need to be explicitly spelled out, including what quality registers and register information from Statistics Sweden. What are the completeness of these data sources? Have the inclusion criteria changed over time? From which register was type 1 diabetes obtained, which diagnosis codes, etc. If the aim is to write a medical paper, then I think the data section should be first in the methods section, and then followed by the statistical methods section. If the aim is to write a methodological paper then the data section can be less.

WE HAVE UPDATED THE DATA SECTION AND MOVED IT TO THE BEGINNING OF THE METHODOLOGY AS SUGGESTED. WE DIDN’T HAVE ACCESS TO THE REGISTERS, RATHER THE NEEDED DATA WERE CURLED OUT FOR US

3. Table 2: Why is the Log-likelihood so much higher for the CPH? Were all models fitted with the same covariates (i.e. same linear predictor of covariate effects), apart from the difference in baseline hazard parameterisation?

YES, WE FITTED THE MODELS WITH SAME COVARIATES, THE HIGHER LIKELIHOOD WAS DUE TO HOW CPH MODEL COMPUTES THE LIKELIHOOD. IT RETURNS PARTIAL LIKELIHOOD. WE HAVE NOTED THIS IN THE MANUSCRIPT.

4. Table 3: The birth cohort effect does not make sense. The FPSR and CPH should yield similar results, if they are proportional hazards models with similar adjustment factors. Please clarify why the models give such different results.

THE FPSR AND CPH WERE MODELED WITH SIMILAR ADJUSTMENT FACTORS. THE ASSUMPTIONS OF THE MODELS COULD HAVE CAUSED THE THIN DIFFERENCES

Minor comments:

5. Abstract: In the results it is stated “hazard” of type 1 diabetes, but are the authors not estimating incidence rates of diabetes? Maybe clarify which disease measure is estimated rather than using the generic term hazard.

WE ESTIMATED THE HAZARD FROM THE MODELS, SO IT IS RIGHT TO USE HAZARD

6. Introd: Can be shortened, and some text around previous literature can be moved to Discussion, I think.

7. Intro row 102: Explain term NPMLE.

IT HAS BEEN EXPLAINED AS NONPARAMETRIC MAXIMUM LIKELIHOOD ESTIMATE

8. Intro row 104: KM curves can be estimated by risk factor groups (covariates), but it is difficult to adjust for multiple confounders simultaneously (however, the curves can be standardized, as a form of adjustment). Standardization is mainly used for one to two variables at the time. Please clarify.

WE HAVE CLARIFIED THIS

9. Does the methods section require all the formula given, or can it be simplified and referenced to original papers instead?

PLOS ONE REQUIRES DETAILED METHODOLOGY, FOR UNDERSTANDING AND REPEATABILITY, HENCE WE RETAIN IT BECAUSE OTHER REVIEWERS ARE HAPPY WITH IT

10. Methods: I don’t understand the “……………………………..(1)” notation in the formulas. Also notation “(i = 1, ………., N)” should perhaps be “(i=1,2,…,N)”. I don’t understand the notation “j= 1; : : : ;P;” or the notation “… … … … … .” or the notation “+ ⋯… … … …+”.

THANK YOU. WE HAVE CORRECTED THESE NOTATIONS AS DIRECTED.

11. Methods row 187: “Odd scale” should be “odds scale” I think?

THANK YOU

12. Methods row 210: The sentence “The position of the internal knots is usually in centiles computed as 100/df.” The placement of knots are at the centiles of event times, I believe.

WE AGREE. THANK YOU. WE HAVE CORRECTED

13. Methods, Dependent variable, row 227: I don’t understand the sentence “The dependent variable is the censored timing (age) of the onset of type-1 diabetes among children. The time was censored on the date of the data collection, emigration and death.” In survival analysis the outcome is two-dimensional and defined by a survival time (with a start and an end), and an event indicator. Please clarify. What does “date of data collection” mean? Is this the date of extraction from the registers?

THANK YOU. WE HAVE CLARIFIED AND CORRECTED THIS.

14. Results: How was the attributable fraction calculated, please explain in the methods section and give a reference.

THANK YOU. WE HAVE ADDED THIS TO THE METHODOLOGY

15. Results, row 270-274: I don’t understand the added value of this section. Why is inference made on crude unadjusted rates? This is better done in adjusted models further down in the results section.

THANK YOU. WE ONLY SHOWED THIS FOR ART VS SPONTANEOUS AND NOT FOR THE ENTIRE COVARIATES. THIS IS NECESSARY FOR READERS TO BE AWARE OF THE DIFFERENCES WITHOUT COVARIATES

16. Results: Row 278: The log rank test is not a test of proportional hazards assumption, I think. Please clarify.

THANK YOU. WE HAVE CORRECTED THIS

6. PLOS authors have the option to publish the peer review history of their article (what does this mean?). If published, this will include your full peer review and any attached files.

Do you want your identity to be public for this peer review? For information about this choice, including consent withdrawal, please see our Privacy Policy.

Reviewer #1: No

Reviewer #2: No

Reviewer #3: No

Reviewer #4: No

---

## [Decision Letter · Decision Letter 1]

16 Mar 2021

PONE-D-21-00472R1

Conception modes and risk of Type-1 diabetes among 1985-2015 Swedish birth cohort: How robust are the survival analysis regression models?

PLOS ONE

Dear Dr. Fagbamigbe,

Thank you for submitting your manuscript to PLOS ONE. After careful consideration, we feel that it has merit but does not fully meet PLOS ONE’s publication criteria as it currently stands. Therefore, we invite you to submit a revised version of the manuscript that addresses the points raised during the review process.

We look forward to receiving your revised manuscript.

Kind regards,

Y Zhan

Academic Editor

PLOS ONE

Reviewers' comments:

Reviewer's Responses to Questions

**Comments to the Author**

1. If the authors have adequately addressed your comments raised in a previous round of review and you feel that this manuscript is now acceptable for publication, you may indicate that here to bypass the “Comments to the Author” section, enter your conflict of interest statement in the “Confidential to Editor” section, and submit your "Accept" recommendation.

Reviewer #1: All comments have been addressed

Reviewer #2: All comments have been addressed

Reviewer #3: (No Response)

2. Is the manuscript technically sound, and do the data support the conclusions?

Reviewer #1: Yes

Reviewer #2: Yes

Reviewer #3: Partly

3. Has the statistical analysis been performed appropriately and rigorously? 

Reviewer #1: Yes

Reviewer #2: Yes

Reviewer #3: No

4. Have the authors made all data underlying the findings in their manuscript fully available?

Reviewer #1: Yes

Reviewer #2: No

Reviewer #3: No

5. Is the manuscript presented in an intelligible fashion and written in standard English?

Reviewer #1: Yes

Reviewer #2: Yes

Reviewer #3: Yes

6. Review Comments to the Author

Reviewer #1: My general comment from the first round of reviews regarding the aim of the study still stands, but if the other reviewers and editors are happy then I have no further comments. The authors have adequately addressed my specific comments.

Reviewer #2: (No Response)

Reviewer #3: Thanks for the opportunity to review this re-submission.

I think the paper has improved in clarity and presentation from the previous submission, but my main concerns have still not been addressed.

Specifically, now that the authors clarified that "This study is a method paper with an application", I think the paper should focus more on the models comparison (which is still just barely discussed):

- The difference in interpretation between accelerated failure time and proportional hazards models is not discussed. Why use one against the other, if they are just a re-parametrisation of each other?

- The differences between the fitted model coefficients (in the application) is not presented, so I am not sure the reader can appreciate what happens when a not-flexible-enough parametric form is assumed. Performing a step-wise procedure with AIC/BIC does not assess robustness, to my eyes;

- Still, there is no much reason to use the flexible parametric models if hazard ratios are the measure of interest here - the Cox model would work just fine, without needing any functional form assumption;

- I still don't think there is strong evidence against proportional hazards. The right-hand-side plot of figure 2 doesn't really show violations of the assumptions, as it is hard to identify non-proportional hazards on the survival scale. Furthermore, if the authors believe there are non-proportional hazards, how are they accommodating that into the analysis?

- It's still not possible to compare a partial likelihood model (the Cox model) with models that are fitted using full likelihood with AIC/BIC, that hasn't been corrected;

- I still don't agree with the conclusion that "the methods of analysis may not be connected with the contradictory findings in earlier studies", we just don't know as we don't see the comparison between all the different models that are being studied here. Further to that, we don't know what the true effect is, so we cannot exclude that both the Cox and the Royston-Parmar models get it wrong, I believe?

In conclusion, I still think the paper needs major modifications if the goal is to study the robustness of different survival regression models. If this is a method paper, I think it should focus more on the methodological part rather than on the application (which I think it's still the case).

7. PLOS authors have the option to publish the peer review history of their article (what does this mean?). If published, this will include your full peer review and any attached files.

Reviewer #1: No

Reviewer #2: No

Reviewer #3: No

---

## [Author Response · Author response to Decision Letter 1]

14 May 2021

Reviewer #1: My general comment from the first round of reviews regarding the aim of the study still stands, but if the other reviewers and editors are happy then I have no further comments. The authors have adequately addressed my specific comments.

Thank you

Reviewer #2: (No Response)

Thank you

Reviewer #3: Thanks for the opportunity to review this re-submission.

I think the paper has improved in clarity and presentation from the previous submission, but my main concerns have still not been addressed.

Specifically, now that the authors clarified that "This study is a method paper with an application", I think the paper should focus more on the models comparison (which is still just barely discussed):

- The difference in interpretation between accelerated failure time and proportional hazards models is not discussed. Why use one against the other, if they are just a re-parametrisation of each other?

Thank you. We have improved on the model comparison, especially in the discussion part as we have already explained the differences between the models in the introduction

- The differences between the fitted model coefficients (in the application) is not presented, so I am not sure the reader can appreciate what happens when a not-flexible-enough parametric form is assumed. Performing a step-wise procedure with AIC/BIC does not assess robustness, to my eyes;

Than you. The fitted model coefficients have been provided

- Still, there is no much reason to use the flexible parametric models if hazard ratios are the measure of interest here - the Cox model would work just fine, without needing any functional form assumption;

Thank you. The study was to compare different models. As you rightly said the Cox compared favourably with the flexible parametric models, although the latter has lower model fit parameters

- I still don't think there is strong evidence against proportional hazards. The right-hand-side plot of figure 2 doesn't really show violations of the assumptions, as it is hard to identify non-proportional hazards on the survival scale. Furthermore, if the authors believe there are non-proportional hazards, how are they accommodating that into the analysis?

We agree, the study was to compare different models. The proportional models compared well with other models. We have reflected these in our results, discussions, findings and abstracts. Nonetheless, the flexible model, as detailed in the literature, has capability of handling non-proportional hazards. As stated in the methodology, we focussed on the hazard scale of the models to ensure that the estimates from the FPSR and CPH models are comparable.

- It's still not possible to compare a partial likelihood model (the Cox model) with models that are fitted using full likelihood with AIC/BIC, that hasn't been corrected;

This has been corrected

- I still don't agree with the conclusion that "the methods of analysis may not be connected with the contradictory findings in earlier studies", we just don't know as we don't see the comparison between all the different models that are being studied here. Further to that, we don't know what the true effect is, so we cannot exclude that both the Cox and the Royston-Parmar models get it wrong, I believe?

We have changed the sentence to read “Hence the methods of analysis may not be disconnected with the contradictory findings in earlier studies”

In conclusion, I still think the paper needs major modifications if the goal is to study the robustness of different survival regression models. If this is a method paper, I think it should focus more on the methodological part rather than on the application (which I think it's still the case).

Thank you for the constructive comments. As you stated, this study is a method paper with an application. So we have discussed the models and the applications.

---

## [Decision Letter · Decision Letter 2]

28 May 2021

PONE-D-21-00472R2

Comparison of the performances of survival analysis regression models for analysis of conception modes and risk of Type-1 diabetes among 1985-2015 Swedish birth cohort

PLOS ONE

Dear Dr. Fagbamigbe,

Thank you for submitting your manuscript to PLOS ONE. After careful consideration, we feel that it has merit but does not fully meet PLOS ONE’s publication criteria as it currently stands. Therefore, we invite you to submit a revised version of the manuscript that addresses the points raised during the review process.

We look forward to receiving your revised manuscript.

Kind regards,

Y Zhan

Academic Editor

PLOS ONE

Journal Requirements:

Reviewers' comments:

Reviewer's Responses to Questions

**Comments to the Author**

1. If the authors have adequately addressed your comments raised in a previous round of review and you feel that this manuscript is now acceptable for publication, you may indicate that here to bypass the “Comments to the Author” section, enter your conflict of interest statement in the “Confidential to Editor” section, and submit your "Accept" recommendation.

Reviewer #3: (No Response)

2. Is the manuscript technically sound, and do the data support the conclusions?

Reviewer #3: Partly

3. Has the statistical analysis been performed appropriately and rigorously? 

Reviewer #3: No

4. Have the authors made all data underlying the findings in their manuscript fully available?

Reviewer #3: No

5. Is the manuscript presented in an intelligible fashion and written in standard English?

Reviewer #3: Yes

6. Review Comments to the Author

Reviewer #3: Thank you for the opportunity to re-review this manuscript.

Overall, I think the focus of the manuscript has greatly improved: I have only a few (relatively) minor comments left to address, which are outlined below.

* "Model selection criteria" section, page 10, robustness of the model wasn't really assessed anywhere, so I would suggest removing that and focussing on fit of the models to the available data.

* "Test of equality of incidence rates of type-1 diabetes" section, page 12, I would suggest reporting the rate difference per 1,000 person-years (or 100,000, as the authors prefer) to improve readability, as the currently reported rates have up to 6 significant digits.

* Again, if the PH assumption is violated, then all PH models yield (possibly) wrong results and non-proportional hazards need to be incorporated in the models. I still don't think Figure 2 shows evident violations of PH, but if the authors believe so, then the analysis (and all the fitted models in the manuscript) need to be updated to reflect this. This is the most important issue that the authors should focus on, in my opinion.

* "Comparison of the models" section, page 14, estimates for PH and AFT models are not directly comparable (as the authors state in the methods section, page 7-8). Therefore, comparing the magnitude of the fitted coefficients does not make sense - this needs to be corrected.

* "Discussion" section, page 18, AIC and BIC for the Cox model are higher because it uses partial likelihood - therefore it's an unfair comparison. This has been fixed elsewhere in the manuscript, but not here; I think it needs to be adjusted here too.

* Line 450, page 19, there's a typo - I think the author references there is Rutherford, not Rutherfold.

* "Strength and limitation" section, I think that (as mentioned before) robustness is not really studied here, so I would reword there.

* "Conclusion" section, please mention that the models performed similarly in this specific setting, but there's no guarantee that this will be the case with other data sources or with other diseases. It's hard to generalise these findings to other settings. It would also be great if the authors could discuss non-poportional hazards: as above-mentioned, they first state that there are non-PH but then use PH (or AFT) models, this needs to be adjusted. A general discussion of non-PH as a strength of FPMs (where it's easy to incorporate this) is also welcome - otherwise, I see no reason to not use the Cox model if relative risk is the main measure of interest and non-PH are ignored.

* Some additional comments: 1) Panel D of Figure 1 is hard to read (and I think the caption "the lines for 'father not diabetic'..." is cut off?), and 2) the names in the first column of figure 5 should be updated to be more descriptive (not just the variable name used in Stata).

7. PLOS authors have the option to publish the peer review history of their article (what does this mean?). If published, this will include your full peer review and any attached files.

Reviewer #3: No

---

## [Author Response · Author response to Decision Letter 2]

1 Jun 2021

Journal Requirements:

Thank you. We have worked on this and ensured that all references are complete and correct. The old references 31 and 53 have been removed (there was no need for replacement) while old references 18, 32,35,36,28 t0 40 and 42 have been updated

6. Review Comments to the Author

Reviewer #3: Thank you for the opportunity to re-review this manuscript.

Overall, I think the focus of the manuscript has greatly improved: I have only a few (relatively) minor comments left to address, which are outlined below.

* "Model selection criteria" section, page 10, robustness of the model wasn't really assessed anywhere, so I would suggest removing that and focussing on fit of the models to the available data.

We have removed the issue about robustness and focussed on fit of the models to the available data.

* "Test of equality of incidence rates of type-1 diabetes" section, page 12, I would suggest reporting the rate difference per 1,000 person-years (or 100,000, as the authors prefer) to improve readability, as the currently reported rates have up to 6 significant digits.

Thank you. We agree with you. We have removed the column with up to 6 digits

* Again, if the PH assumption is violated, then all PH models yield (possibly) wrong results and non-proportional hazards need to be incorporated in the models. I still don't think Figure 2 shows evident violations of PH, but if the authors believe so, then the analysis (and all the fitted models in the manuscript) need to be updated to reflect this. This is the most important issue that the authors should focus on, in my opinion.

Thank you for pointing this out. We have now updated our text to indicate that PH assumptions were not violated

* "Comparison of the models" section, page 14, estimates for PH and AFT models are not directly comparable (as the authors state in the methods section, page 7-8). Therefore, comparing the magnitude of the fitted coefficients does not make sense - this needs to be corrected.

We agree totally. We have corrected this statement

* "Discussion" section, page 18, AIC and BIC for the Cox model are higher because it uses partial likelihood - therefore it's an unfair comparison. This has been fixed elsewhere in the manuscript, but not here; I think it needs to be adjusted here too.

Thank you. We have fixed the sentence 

* Line 450, page 19, there's a typo - I think the author references there is Rutherford, not Rutherfold.

Thank you. We have corrected this

* "Strength and limitation" section, I think that (as mentioned before) robustness is not really studied here, so I would reword there.

We appreciate this concern. Yes, we have removed robustness across the manuscript and replaced with model fit

* "Conclusion" section, please mention that the models performed similarly in this specific setting, but there's no guarantee that this will be the case with other data sources or with other diseases. It's hard to generalise these findings to other settings. It would also be great if the authors could discuss non-poportional hazards: as above-mentioned, they first state that there are non-PH but then use PH (or AFT) models, this needs to be adjusted. A general discussion of non-PH as a strength of FPMs (where it's easy to incorporate this) is also welcome - otherwise, I see no reason to not use the Cox model if relative risk is the main measure of interest and non-PH are ignored.

Thank you for this suggestion. It has been incorporated appropriately in the discussion and conclusion sections

* Some additional comments: 1) Panel D of Figure 1 is hard to read (and I think the caption "the lines for 'father not diabetic'..." is cut off?), and 2) the names in the first column of figure 5 should be updated to be more descriptive (not just the variable name used in Stata).

 Thank you for the constructive comments. We have fixed this. Actually, the lines for father not diabetic and mother not diabetic overlapped.

Also, we have reproduced the forest plots to ensure easy reading.

---

## [Editor Report · Decision Letter 3]

4 Jun 2021

Comparison of the performances of survival analysis regression models for analysis of conception modes and risk of Type-1 diabetes among 1985-2015 Swedish birth cohort

PONE-D-21-00472R3

Dear Dr. Fagbamigbe,

We’re pleased to inform you that your manuscript has been judged scientifically suitable for publication and will be formally accepted for publication once it meets all outstanding technical requirements.

Kind regards,

Y Zhan

Academic Editor

PLOS ONE
---

## [Editor Report · Acceptance letter]

18 Jun 2021

PONE-D-21-00472R3 

Comparison of the performances of survival analysis regression models for analysis of conception modes and risk of Type-1 diabetes among 1985-2015 Swedish birth cohort 

Dear Dr. Fagbamigbe:

I'm pleased to inform you that your manuscript has been deemed suitable for publication in PLOS ONE. Congratulations! Your manuscript is now with our production department. 

Kind regards, 

on behalf of

Dr. Y Zhan 

Academic Editor

PLOS ONE